# Chest ImaGenome Dataset for Clinical Reasoning

**Joy T. Wu[1], Nkechinyere N. Agu[2], Ismini Lourentzou[3], Arjun Sharma[4], Joseph A. Paguio[5], Jasper S. Yao[5], Edward C. Dee[6], William Mitchell[4], Satyananda Kashyap[1], Andrea Giovannini[1], Leo A. Celi[4], Mehdi Moradi[1]**

[1]IBM Almaden Research Center, San Jose, CA 95120, USA
[2]Rensselaer Polytechnic Institute, Troy, NY 12180, USA
[3]Virginia Polytechnic Institute and State University, Blacksburg, VA 24061, USA
[4]MIT Critical Data, Cambridge, MA 02139, USA
[5]Albert Einstein Healthcare Network-Philadelphia Campus, PA 19141, USA
[6]Harvard Medical School, Boston, MA 02115, USA

## Abstract

Despite the progress in automatic detection of radiologic findings from chest X-ray (CXR) images in recent years, a quantitative evaluation of the explainability of these models is hampered by the lack of locally labeled datasets for different findings. With the exception of a few expert-labeled small-scale datasets for specific findings, such as pneumonia and pneumothorax, most of the CXR deep learning models to date are trained on global "weak" labels extracted from text reports, or trained via a joint image and unstructured text learning strategy. Inspired by the Visual Genome effort in the computer vision community, we constructed the first Chest ImaGenome dataset with a scene graph data structure to describe $242,072$ images. Local annotations are automatically produced using a joint rule-based natural language processing (NLP) and atlas-based bounding box detection pipeline. Through a radiologist constructed CXR ontology, the annotations for each CXR are connected as an anatomy-centered scene graph, useful for image-level reasoning and multimodal fusion applications. Overall, we provide: i) $1,256$ combinations of relation annotations between 29 CXR anatomical locations (objects with bounding box coordinates) and their attributes, structured as a scene graph per image, ii) over $670,000$ localized comparison relations (for improved, worsened, or no change) between the anatomical locations across sequential exams, as well as ii) a manually annotated gold standard scene graph dataset from $500$ unique patients.

## Introduction

Chest X-rays (CXR) are among the commonly ordered radiology exams, mostly for screening but also for diagnostic purposes. Recently, multiple large CXR imaging datasets have been released by the research community [1, 2, 3, 4]. These can be used to develop automatic abnormality detection or report generation algorithms. For detecting specific abnormalities from images, natural language processing (NLP) algorithms have been used to extract "weak" global image-level labels (CXR abnormalities) from the associated CXR reports [4, 5, 6, 7]. For automatic report generation, self-supervised joint text and image architectures [8, 9, 10, 11, 12], first inspired by the image captioning related work in the non-medical domain [13, 14, 15, 16, 17], have been used to produce preliminary free-text radiology reports. However, both approaches lack rigorous localization assessment for explainability, namely whether the model attended to the relevant anatomical location(s) for predictions. This missing feature is critical for clinical applications. The joint image and text learning strategy are also known to learn heavy language priors from the text reports without having learned to interpret the

imaging features [18, 19]. Furthermore, even though architectures suitable for comparing imaging changes are available [20, 21], limited work has focused on automatically deriving comparison relations between exams from large datasets for the purpose of training imaging models that can track progress for a wide variety of CXR findings or diseases.

To the best of our knowledge, no prior work in CXR has attempted to automatically extract relations between CXR attributes (labels) from reports and their anatomical locations (objects with bounding box coordinates) on the images as documented by the reporting radiologists, nor has there been any localized relation annotations between sequential CXR exams. Research on these two topics is valuable because radiology reports in effect are records of radiologists' complex clinical reasoning processes, where the anatomical location of observed imaging abnormalities is often used to narrow down on potential diagnoses, as well as for integrating information from other clinical modalities (e.g. CT findings, labs, etc) at the anatomical levels. Sequential exams are also routinely used by bedside clinicians to track patients' clinical progress after being started on different management paths. Therefore, documentations comparing sequential exams are prevalent in CXR reports and are clinically meaningful relations to learn about. Automatically extracting radiology knowledge graphs and disease progression information from reports will help improve explainability evaluation and widen downstream clinical applications for CXR imaging algorithm development.

Many algorithms for object detection and domain-knowledge-driven reasoning require a starting dataset that has localized labels on the images and meaningful relationships between them. In the non-medical domain, large locally labeled graph datasets (e.g., Visual Genome dataset [22]) have enabled the development of algorithms that can integrate both visual and textual information and derive relationships between observed objects in images [23, 24, 25]. In addition, they have spurred a whole domain of research in visual question answering (VQA) and visual dialogue (VD), with the aim of developing interactive AI algorithms capable of reasoning over information from multiple sources [26, 27, 28]. These location, relation and semantics aware systems aim to capture important elements in image data in relation to complex human languages, in order to conversationally interact with humans about the visual content. In the medical domain, such systems may help with automatic image and text information retrieval tasks from databases or improve end-user trust by allowing clinicians to interactively question trained models to assess the consistency of predictions.

In this paper, we present the Chest ImaGenome dataset, a large multi-modal (text and images) chronologically ordered scene graph dataset for frontal chest x-ray (CXR) images. This dataset is an important step towards addressing the missing link of large locally labeled graph datasets in the medical imaging domain. The goal for releasing this dataset is to spur the development of algorithms that more closely reflect radiology experts' reasoning processes. In addition, automatically describing localized imaging features in recognized medical semantics is the first step towards connecting potentially predictive pixel-level features from medical images with the rest of the digital patient records and external medical ontologies. These connections could aid both the development of anatomically relevant multi-modal fusion models and the discovery of localized imaging fingerprints, i.e., patterns predictive of patient outcomes. Through **PhysioNet's credentialed access** (see **license**), we make the first Visual Genome-like graph dataset in the CXR domain accessible for the research community.

**Related work**: A few CXR datasets have localized abnormality annotations [29, 30, 31] that are curated manually. These are high quality gold standard ground truth datasets but tend to be smaller in scale (< 30,000 images) and have a narrow coverage, with typically only 1-2 labels. In addition, since most labeling efforts only have abnormality semantics attached, no direct relationships with the affected anatomical locations are available.

Two recent CXR datasets have labels for anatomies described in the reports. In [32], a small manually annotated dataset (2000 reports) included 10 abnormalities that are individually associated with 29 unique spatial locations (anatomies) at the report level. Another CXR dataset has automatically extracted abnormality and anatomy labels as disconnected concepts that are only correlated at the study level from 160,000 reports using a supervised NLP algorithm [7]. This was trained on a smaller set of manually annotated data. Neither datasets contain localized annotations for the associated CXR images, nor any comparison relation annotations between sequential exams, both of which are available in the Chest ImaGenome dataset. In Table 1, we present a comparison of our Chest ImagGenome dataset with other datasets available in the literature.

Table 1: Summary of existing chest X-ray datasets

| Dataset | Annotation Level | Annotation Method | Num Labels | Anatomy Labeled | Graph | Dataset Size | Temporal Labels | Reports |
|---|---|---|---|---|---|---|---|---|
| SIIM-ACR Pneumothorax Segmentation [30] | Segmentation | Manual + augmented | 1 | No | No | 12,047 | No | No |
| RSNA Pneumonia Detection Challenge [29] | Bounding Boxes | Manual | 1 | No | No | 30,000 | No | No |
| Indiana University Chest X-ray collection [2] | Global | Automated | 10 | No | No | 3,813 | No | Yes |
| NIH CXR dataset [3] | Global | Automated | 14 | No | No | 112,120 | No | No |
| PLCO [33] | Global | Automated | 24 | Yes | No | 236,000 | Yes | No |
| Stanford CheXpert [4] | Global | Automated | 14 | No | No | 224,316 | No | No |
| MIMIC-CXR [1] | Global | Automated | 14 | No | No | 377,110 | No | Yes |
| Dutta [32] | Global | Manual | 10 | Yes | Yes | 2,000 | No | Yes |
| PadChest [7] | Global | Manual + automated | 297 | Yes | No | 160,868 | No | Yes |
| Montgomery County Chest X-ray [31] | Segmentation | Manual | 1 | Yes | No | 138 | No | No |
| Shenzen Hospital Chest X-ray [31] | Segmentation | Manual | 1 | Yes | No | 662 | No | No |
| **Chest ImaGenome** | Bounding Boxes | Automated | 131 | Yes | Yes | 242,072 | Yes | Yes |

## Methods

The Chest ImaGenome dataset was derived from the MIMIC-CXR dataset [1], which has been de-identified. This derived dataset retains the added annotations and the source image tags but not the CXR images, which users are expected to separately download from the **MIMIC-CXR database**. The institutional review boards of the Massachusetts Institute of Technology (No. 0403000206) and Beth Israel Deaconess Medical Center (BIDMC)(2001-P-001699/14) both approved the use of the MIMIC database for research. All authors working with the data have individually completed required HIPPA training and been granted data access approval from PhysioNet.

**Silver Dataset Construction**

The Chest ImaGenome dataset construction is inspired by the Visual Genome dataset [22]. Whereas Visual Genome utilized web-based and crowd-sourced methods to manually collect annotations, the Chest ImaGenome harnessed NLP, a CXR ontology, and image segmentation techniques to automatically structure and add value to existing CXR images and their free-text reports, which were collected from radiologists in their routine workflow. We used atlas-based bounding box extraction techniques to structure the anatomies on $242,072$ frontal CXR images, anteroposterior (AP) or posteroanterior (PA) view, and used a rule-based text-analysis pipeline to relate the anatomies to various CXR attributes (finding, diseases, technical assessment, devices, etc) extracted from $217,013$ reports. Altogether, we automatically annotated $242,072$ scene graphs that locally and graphically describe the frontal images associated with these reports (one report can have one or more frontal images). Our goal is to not only locally label attributes relevant for key anatomical locations on the CXR images, but also to extract documented radiology knowledge from a large corpus of CXR reports to aid future semantics-driven and multi-modal clinical reasoning works.

Table 2 describes the parallels between the Chest ImaGenome and Visual Genome datasets. The key differences are in the construction methodology, the currently much smaller range of possible objects and attributes (due to having only the CXR imaging modality), and the introduction of comparison relations between sequential images in the Chest ImaGenome dataset. We define the nodes and edges in the graph (Supplementary Table 6) based on clinical relevance and resources in the context for medical imaging exams like CXRs. In addition, two **key assumptions** are made in the construction of the Chest ImaGenome dataset:

1) CXR imaging observations can be normalized to relationships between the visualized anatomical locations (object nodes) and the abnormalities, devices or other CXR descriptions (attribute nodes) that the locations contain. Thus, the variety of detected objects is confined by the granularity of anatomical location detection on images and from reports.

2) The exam timestamps in the original MIMIC-CXR dataset can be used to chronologically order the CXR exams from the same patient within the original MIMIC CXR dataset's collection period and there are minimal missing exams for each patient. This is based on discussions with the MIMIC team and MIMIC-CXR's documented data collection strategy. The original data curators included all CXR exams in the radiology imaging archives for patients who were at any time point admitted to the BIDMC's Emergency Department within a continuous 2-year-period. Therefore, we related any comparison descriptions (normalized to 'improved', 'worsened' and 'no change') of attribute(s) in different anatomical location(s) to the same anatomical location(s) on the exam image(s) immediately before the current exam. Clinically, the extracted comparison relations are intended to allow longitudinal modeling of disease progression for different CXR anatomies.

The construction of the Chest ImaGenome dataset builds on the works of [5, 36]. In summary, the text pipeline [5] first sections the report and retains only the finding and impression sentences, and

Table 2: Parallels between the Chest ImaGenome and Visual Genome datasets.

| Element | Chest ImaGenome | Visual Genome |
|---|---|---|
| Scene | One frontal CXR image in the current dataset. | One (non-medical) everyday life image. |
| Questions | For now, there is only one question per CXR, which is taken from the patient history (i.e., reason for exam) section from each CXR report. | One or more questions that the crowd source annotators decided to ask about the image where the information from each question and the image should allow another annotator to answer it. |
| Answers | N/A currently. However, report sentences are biased towards answering the question asked in the reason for exam sentence;hence, the knowledge graph we extract from each report should contain the answer(s). | This was collected as answer(s) to the corresponding question(s) asked of the image. |
| Sentences (Region descriptions) | Sentences from the finding and impression sections of a CXR report describing the exam as collected from radiologists in their routine radiology workflow. | True natural language descriptive sentences about the image collected from crowd-sourced everyday annotators. |
| Objects (nodes) | Anatomical structures or locations that have bounding box coordinates on the associated CXR image, and is indexed to the UMLS ontology [34]. | The people and physical objects with bounding box coordinates on the image and indexed to WordNet ontology [35]. |
| Attributes (nodes) | Descriptions that are true for different anatomical structures visualized on the CXR image (e.g., There is a right upper lung [object] opacity [attribute]), indexed to the UMLS ontology [34]. No Bbox coordinates. | Various descriptive properties of the objects in the image (e.g., The shirt [object] is blue [attribute]), indexed to WordNet ontology [35]. No Bbox coordinates. |
| Relations: object and attribute | The relationship(s) between an anatomical object and its attribute(s) from the same CXR image (e.g., There is a [relation] right upper lung [object] opacity [attribute]). | The relationship(s) between an object and its attribute(s) from the same image ( e.g., The shirt [object] is [relation] blue [attribute]). |
| Relations: object and object | The comparison relationship (index to UMLS [34]) between the same anatomical object from two sequential CXR images for the same patient (e.g., There is a new [relation] right lower lobe [current and previous anatomical objects] atelectasis [attribute]). | The relationship (indexed to WordNet [35]) between objects in the same image (e.g., The boy [object 1] is beside [relation] the bus [object 2]). |
| Relations: parent and child | To make the graph for each image logically consistent and correct as learnable and consumable radiology knowledge, affirmed parent-child relations between nodes are embedded in the scene graphs – i.e., if a child attribute is related to an object, then its parent would be too (e.g., if right lung has consolidation [child], then it also has lung opacity [parent]). | N/A due to different graph construction strategy and goals. The annotators were asked to describe any (but not all) relations they observe in an image. |
| Scene graph | Constructed from the objects, the attributes and the relationships between them for the image. | Same but the nodes and edges overall would be more varied than Chest ImaGenome for now. |
| Sequence* | A super-graph for a set of chronologically ordered series of exams for the same patient. | N/A, but would be a graph for a video in the non-medical context. |

then utilizes a CXR concept dictionary (lexicons) to spot and detect the context (negated or affirmed) of 271 different CXR related named-entities from each retained sentence. The lexicons were curated in advance by two radiologists in consensus using a concept expansion and vocabulary grouping engine [37]. A set of sentence-level filtering rules are applied to disambiguate some of the target concepts (e.g., 'collapse' mention in CXR report can be about lung 'collapse' or related to spinal fracture as in vertebral body 'collapse'). Then the named-entities for CXR labels (attributes) are associated with the name-entities for anatomical location(s) described in the same sentence with a SpaCy natural language parser [38].

Using a CXR ontology constructed by radiologists, a scene graph assembly pipeline corrected obvious attribute-to-anatomy assignment errors (e.g., lung opacity wrongly assigned to mediastinum). Finally,

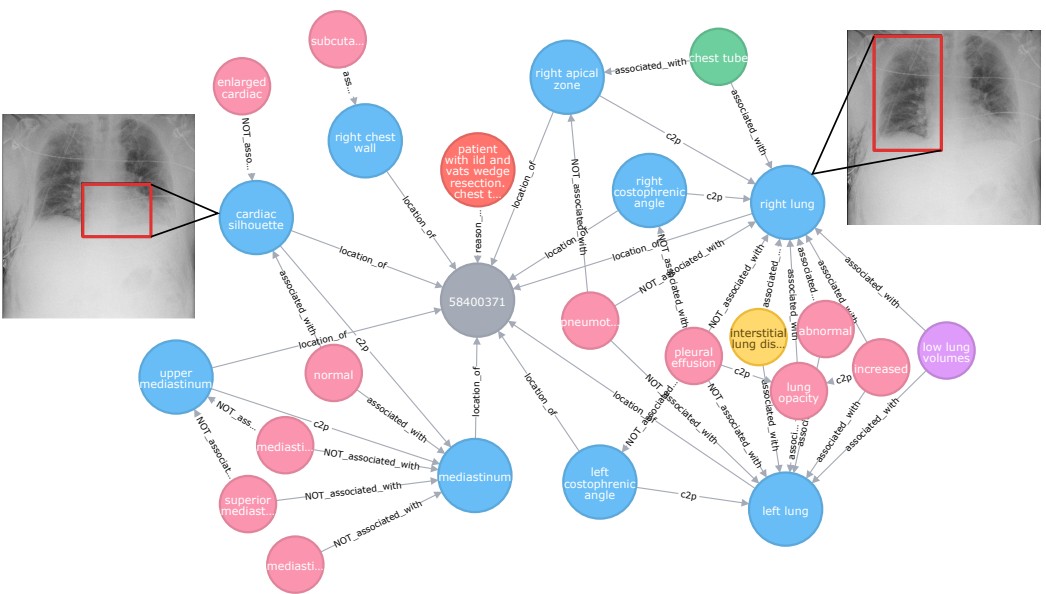

Figure 1: A radiology knowledge graph extracted for one CXR report (grey), with patient history from indication for exam (orange), anatomical locations (blue) and their associated attributes, including anatomical findings (pink), diseases (yellow), technical assessment (purple) and devices (green) nodes. The blue anatomy nodes (a.k.a. objects) also have corresponding bounding box coordinates on the CXR image, which are shown for two examples.

the attributes for each of the target anatomical regions from repeated sentences are grouped to the exam level. The result is that, from each CXR report, we extract a radiology knowledge graph where CXR anatomical locations are related to different documented CXR attribute(s). The "reason for exam" sentence(s) from each report, which contain free text information about prior patient history, are separately kept in the final scene graph JSONs. Patient history information is critical for clinical reasoning but is a piece of information that is not technically part of the "scene" for each CXR.

For detecting the anatomical "objects" on the CXR images that are associated with the extracted report knowledge graph, a separate anatomy atlas-based bounding box pipeline extracts the coordinates of those anatomies from each frontal image. This pipeline is an extension of prior work that covers additional anatomical locations in this dataset [36]. In addition, we manually validated or corrected the bounding boxes for 1, 071 CXR images (with and without disease, and excluded gold standard subjects) to train a Faster-RCNN CXR bounding box detection model, which we used to correct failed bounding boxes (too small or missing) from the initial bounding box extraction pipeline ( 7%). Finally, for quality assurance, we manually annotated 303 images that had missing bounding boxes for key CXR anatomies (lungs and mediastinum).

Extracting comparison relations between sequential exams at the anatomical level is another goal for the Chest ImaGenome dataset. After checking with the MIMIC team and reviewing their dataset documentation, we assume that the timestamps in the original MIMIC-CXR dataset can be used to chronologically order the exams for each patient. We then correlated all report descriptions of changes (grouped as improved, worsened, or no change) between sequential exams with the anatomical locations described at the sentence level. To extract these comparison descriptions, we used a concept expansion engine [37] to curate and group relevant comparison vocabularies used in CXR reports. These comparison relations extracted between anatomical locations from sequential CXRs are only added to the final scene graphs for every patient's second or later CXR exam(s), i.e., comparison relations described in the first study of each patient in the MIMIC-CXR dataset are not added to the Chest ImaGenome dataset.

Finally, we have mapped all object and attribute nodes and comparison relations in the dataset to a Concept Unique Identifier (CUI) in the Unified Medical Language System (UMLS) [34]. The UMLS ontology has incorporated the concepts from the Radlex ontology [31], which targets the radiology

domain. Choosing UMLS to index the Chest ImaGenome dataset widens its future applications in clinical reasoning tasks, which would invariably require medical concepts and relations outside the radiology domain. An example of a CXR scene graph is shown in Figure 1.

**Gold Standard Dataset Collection**

In collaboration with clinicians (radiology and internal medicine M.D.'s) from multiple academic institutions, we curated a dual validated gold standard dataset to 1) evaluate the quality of the silver Chest ImaGenome dataset we automatically generated, and 2) to serve as a benchmark resource for future research using the dataset. Due to resource constraints, we created the gold standard dataset using a validation plus correction strategy. We randomly sampled 500 unique patients from the Chest ImaGenome dataset that had two or more sequential CXR exams. Overall, we targeted three aspects of the scene graph dataset generation process to evaluate separately: A) the object-to-attribute relations (i.e., CXR knowledge graph) extracted from individual reports, B) the object-to-object comparison relations extracted between sequential CXR reports, and C) the anatomical location detection (i.e., the bounding box extraction pipeline) for the CXR images. For details about the gold standard dataset annotation process, see Supplementary (Section C).

# Data description

The Chest ImaGenome dataset is committed to the PhysioNet repository in two main directories, one for the scene graphs that are automatically generated ("silver_dataset"), and another for the 500 unique patient subset that was manually validated and corrected ("gold_dataset"). Overall, $242,072$ scene graphs were automatically derived from $217,013$ unique CXR studies. The nodes and edges in the graph are defined in detail in Supplementary Table 6. On average 7 anatomical objects and 5 attributes are extracted from each study report. However, up to 29 anatomy objects can be detected in each CXR image with a percentage of misses $< 0.02\%$ for most objects (See Table 7 in Supplementary material). In addition, even without considering the related attribute(s), $678,543$ object-object comparison relations are extracted between anatomies across $128,468$ pairs of sequential CXR images. Detailed dataset characteristics are explained and provided in the PhysioNet repository (generate_scenegraph_statistics.ipynb). Figure 2 shows an example of all the anatomical bounding boxes.

**Chest ImaGenome Scene Graph JSONs**

The 'silver_dataset/scene_graph.zip' file is a directory that contains multiple JSON files, one for each scene graph. Each scene graph describes one frontal chest X-ray image. The structure for each scene graph JSON is described by components for easier explanation in Supplementary (Section B). The first level of the JSON in Supplementary (B.1) describes the patient or study level information that may not be available in the image. The fields are: 'image_id' (dicom_id in MIMIC-CXR), 'viewpoint' (AP or PA), 'patient_id' (subject_id in MIMIC-CXR), 'study_id' (study_id in MIMIC-CXR), 'gender' and 'age_decile' demographics (from MIMIC-CXR's metadata), 'reason for exam' (patient history sentence(s) from the CXR reports with age removed), 'StudyOrder' (the order of the CXR study for the patient, which is derived from chronologically ordering the DICOM timestamps), and 'StudyDateTime; (from MIMIC's dicom metadata, which had been de-identified into the future).

For each scene graph, there are 3 separate nested fields to describe the "objects" on the CXR images, the "attributes" related to the different objects as extracted from the corresponding reports, and "relationships" to describe comparison relations between sequential CXR images for the same patient. These 3 fields are a list of dictionaries, where the format of each dictionary is modeled after the respective JSONs in the Visual Genome dataset [22].

For objects, each dictionary has the format shown in Supplementary (B.2). The 'object_id' is unique across the whole dataset for the anatomical location on the particular image. Fields 'x1', 'y1', 'x2', 'y2', 'width' and 'height' are for a padded and resized 224x224 CXR frontal image, where coordinates 'x1', 'y1' are for the top left corner of the bounding box and 'x2', 'y2' are for the bottom right corner. The bounding box coordinates in the original image are denoted with 'original_*'. The remaining fields: 'bbox_name' is the name given to the anatomical location within the Chest ImaGenome dataset, and is useful for lookups in other parts of the scene graph JSON; 'synsets' contain the UMLS CUI for the anatomical location concept; and the 'name' is the UMLS name for that CUI [34]. Note

224 that CXRs are 2D images of a 3D structure so there are many overlying anatomical locations. A
225 sample of 17 of the anatomical objects is plotted on a CXR as shown in Figure 2.

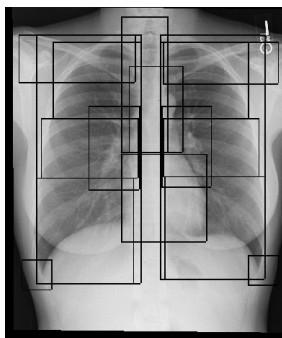

Figure 2: Sample CXR case with 17 overlaying clavicles, lung and mediastinum related anatomical bounding boxes (objects).

226 Each attribute dictionary, e.g., Supplementary (B.3), aims to summarize all the CXR attribute de-
227 scriptions for one anatomical location ('bbox_name'). This means, for a particular CXR anatomical
228 location, all the sentences describing attributes related to it have been grouped into the 'phrases' field,
229 where the order of sentences in the original report has been maintained. However, an anatomical
230 location may not always be described or implied in the report. In that case, looking up dictio-
231 nary['bbox_name'] will be False. The fields 'synsets' and 'name' are the same as in the objects'
232 dictionaries, where they describe the UMLS CUI information for the anatomical location concept.

233 The 'attributes' field contains the relations between the anatomical location and the CXR attributes
234 extracted from the respective sentences. Note that there can be multiple attributes extracted from
235 each sentence. Therefore, the 'attributes' field is a list of lists. The 'attributes' in the lists follow
236 the pattern of < categoryID | relation | label_name >, where 'categoryID' is the radiology semantic
237 category the authors gave to the CXR concept in consultation with multiple radiologists, and relation
238 is the NLP context relating the label_name to the anatomical location as an attribute. If the relation is
239 'no', then the 'label_name' is specifically negated in the sentence. If the relation is 'yes', then the
240 'label_name' is affirmed in the sentence. The order of the lists in the 'attribute_ids' field follow the
241 lists in the 'attributes' field and map each 'label_name' to UMLS CUIs. Thus, the way the Chest
242 ImaGenome dataset is formulated, one can interpret a statement such as the 'right lung' <has no>
243 'lung opacity' as true in the extracted radiology knowledge graph, whereby each node has been
244 mapped to an externally recognized ontology.

245 The certainty of each relation in the CXR knowledge graph can be optionally further modified by the
246 cues from the 'severity_cues' and 'temporal_cues' fields in each attribute dictionary. The severity
247 cues can include 'hedge', 'mild', 'moderate' or 'severe', which are only assigned by co-occurrence
248 at the sentence level. These extractions can benefit from future NLP improvement. Similarly, the
249 temporal cues can modify the relation as either 'acute' or 'chronic' depending on clinical use cases.

250 The Chest ImaGenome categoryIDs can be used to differentiate the use case for different attributes:

251 • **anatomicalfinding** - findings of anatomies where there is some subjectivity in the grouping of the
252 phrases used to extract the labels.

253 • **disease** - descriptions that are more diagnostic level and often require patient information outside
254 the image and most subjective to the reading radiologist's inference/impression.

255 • **nlp** - normal / abnormal descriptions about different anatomical locations and can be subjective.

256 • **technicalassessment** - image quality issues affecting interpretation of CXR observations.

257 • **tubesandlines** - medical support devices where radiologists need to report any placement issues.

258 • **devices**: medical devices where placement issues are less relevant

259 • **texture** - these are only present in the 'texture_cues' field, we kept a set of highly non-specific
260 attributes (e.g. opacity, lucency, interstitial, airspace) that tend to form the initial most objective
261 descriptions about what is observed in the images by radiologists.

262 Finally, for comparison relationships, each dictionary has the format shown in Supplementary (B.4).
263 Each relationship dictionary describes the comparison relation(s) relevant for only one anatomical

location ('bbox_name'). The 'relationship_id' uniquely identifies each comparison relationship between the object ('subject_id') on the current exam and the object ('object_id' for the same anatomical location) from the previous exam. The 'predicate' and 'synsets' are the UMLS CUIs for 'relationship_names', which is a list with usually one (but could be more) comparison relation type, which can be in ['comparison|yes|improved', 'comparison|yes|worsened', 'comparison|yes|no change']. The 'attributes' field records the attributes that are related to the anatomical location as per the sentence from the original report (kept in the 'phrase' field) that describes the comparison relationship.

**CXR Scene Graphs Rendered in an Enriched RDF Format**

Supplementary (B.5): Radiology report sentences are fairly repetitive. Therefore, in the scene graph JSONS, one could see similar information described multiple times in different sentences for a study. In addition, in the MIMIC reports we worked with, each report could also have a preliminary read section (recorded by trainee radiologists - i.e., resident M.D.s) that comes before the final report section (approved by a fully trained and experienced radiologist). Therefore, occasionally, the extraction from the sentences near the beginning of a CXR report can be different from the conclusion sentences later in the report. To render the scene graphs easier for downstream utilization, we also provide post-processing utils (scenegraph_postprocessing.py) to roll the annotations up to the study level for each relation. This is done by taking the last relation extracted for each anatomical location and attribute combinations for a report. The processing utils can either render the scene graphs in a tabular format or represent the information in a simpler enriched RDF format, which we used to generate the graph visualizations in Figure 1.

**Gold Standard Dataset Tables**

We curated a manual gold standard evaluation dataset to measure the quality of the automatically derived annotations in the Chest ImaGenome dataset and for model benchmarking. Here we describe the three gold standard ground truth files in the "gold_dataset" directory. They are in tabular format for ease of comparison purposes.

• *gold_attributes_relations_500pts_500studies1st.txt* is the ground truth file which contains 21,594 object-to-attribute relations manually annotated for 3,042 sentences from the *first* CXR study for 500 unique patients. The notebook 'object-attribute-relation_evaluation.ipynb' explains in detail how we it to calculate the performance of object-to-attribute relation extraction.

• *gold_comparison_relations_500pts_500studies2nd.txt* is the ground truth file which contains 5,156 object-object (per attribute) comparison relations for 638 sentences from the *second* CXR study for the same 500 unique patients. The notebook 'object-object-comparison-relation_evaluation.ipynb' uses it to calculate the performance for object-to-object-comparison relation extraction.

• The four *bbox_coordinate_annotations\*.csv* files contain the manually annotated bounding box coordinates for the objects on the corresponding 1,000 unique CXR images. The notebook 'object-bbox-coordinates_evaluation.ipynb' calculates the bounding box object detection performance using these ground truth files.

• Lastly, *final_merging_report_and_bbox_ground_truth.ipynb* combines the manual text and anatomical bbox annotations as *gold_object_attribute_with_coordinates.txt* and *gold_object_comparison_with_coordinates.txt*.

Additional supporting files for measuring the performance of the silver dataset against the gold standard are described in Supplementary (Section D):

# Dataset Evaluation

Table 3 ('analysis/generated via object-attribute-relation_evaluation.ipynb') reports the NLP pipeline's precision, recall and F1 scores for extracting the relationships between objects (anatomical locations) and CXR attributes (findings, diseases, technical assessment, etc) in the scene graphs. Since at their most granular level, the annotations are at the sentence-level, we report both the sentence-level and report-level results for 500 reports from the first exam of each patient. However, for most purposes, report-level annotations (the last annotation for each object-attribute relation for a study) are most suitable for downstream uses. The majority of the false positive results are due to failure to detect

| Metric | Sentence-level | Report-level |
| --- | --- | --- |
| # of annotations | 21593 | 16569 |
| Precision | 0.932 | 0.938 |
| Recall | 0.945 | 0.939 |
| F1-score | 0.939 | 0.939 |

Table 3: Object-attribute relations. Estimated inter-annotator (IA) agreement on 500 reports from first study: 0.984.

| Metric | Sentence-level | Report-level |
| --- | --- | --- |
| # of annotations | 5154 / 1787 | 3993 / 1374 |
| Precision | 0.831 / 0.856 | 0.832 / 0.858 |
| Recall | 0.590 / 0.663 | 0.762 / 0.790 |
| F1-score | 0.690 / 0.747 | 0.796 / 0.823 |

Table 4: Object-object comparison relations (attribute-sensitive / attribute-blind). IA on 500 reports from second study: 0.962.

the laterality (i.e., left v.s. right) of attributes correctly as this information can often cross sentence boundaries, which is beyond the current NLP pipeline.

Table 4 (generated via 'analysis/object-object-comparison-relation_evaluation.ipynb') shows the NLP results for comparison relations (improved, worsened, no change) between various anatomical locations described for the current study as compared to the patient's previous study. The results are again shown at both sentence-level and report-level for 500 reports from the second exam of each patient. For the attribute-sensitive results, a relation is correct if it describes the correct comparison and attribute for an object. Attribute-blind relations are correct as long as the object-to-object comparison relation is correct. Since comparison relations can cross both sentence and report boundaries, the performance from the current per sentence-based NLP pipeline is lower.

Lastly, Table 7 in Supplementary shows more detailed evaluation at the object-level (anatomical location). The F1 scores are calculated for relations extracted between objects and attributes from the 500 gold standard reports (first study), which is a breakdown of report-level results in Table 3 for the bounding boxes (Bboxes) shown. Using the $1,000$ CXR images in the gold standard dataset, we also calculated the intersection over union (IoU) between the automatically extracted Bboxes and the validated and corrected Bboxes (analysis/object-bbox-coordinates_evaluation.ipynb). Since we used an agree-or-correct annotation strategy for more efficient annotation, we also show the percentage of bounding boxes requiring manual correction in the gold dataset and the percentage missing in the final Chest ImaGenome dataset. Missing bounding boxes could be due to Bbox extraction failure or the anatomical location genuinely not being visible in the image (i.e., cut off or not in field of view), which is not uncommon for the costophrenic angles and apical zones. Per attribute level performance is available on the PhysioNet repository ('analysis/affirmed_attributes_eval4paper.csv').

## Clinical Applications

There are numerous clinical topics that may be explored for a dataset that links anatomic structures with individual abnormalities and simultaneously provides comparison relation annotations for sequential images. Monitoring the progression of pathologies that are visualized through chest imaging is the most unexplored clinical application of this dataset. In the in-patient setting, diagnosis and monitoring of pneumonia are typically performed through comparisons of sequential CXR images from admission[39]. The same management principle may apply to the evaluation of the progression of other diseases, such as pneumothorax, pulmonary edema, acute respiratory distress syndrome, or congestive heart failure [40, 41, 42]. In the outpatient setting, surveillance of incidental pulmonary nodules, malignancies, tuberculosis, or interstitial lung disease is done through chest imaging in several-month intervals [43, 44, 45, 46]. Furthermore, the methodological concepts of this dataset could be extended to other modes of imaging, such as computed tomography (CT), and magnetic resonance (MR) imaging, etc, further expanding the potential clinical utility of this project.

**Consistent dataset splits for performance reporting**: For reproducibility, we include splits for train, valid and test sets in the "silver_dataset/splits" directory. The random data split was done at the patient level. We also included a file (images_to_avoid.csv) with image IDs ('dicom_id') and 'study_id's for patients in the gold standard dataset, which should all be excluded from training and validation.

As described, Chest ImaGenome has been constructed with multiple possible downstream tasks in mind. Here, we showcase two example tasks that can have the most immediate clinical applications, (i) outputting both the location and the type of CXR attribute for an image (Example Task 2) and (ii) comparing whether a location has worsened or improved across sequential exams (Example Task 1). Clinically, the two chosen types of tasks are the two most important ones for radiologists to report when interpreting CXRs.

Table 5: Anatomically localized CXR attribute detection (AUC scores). L1: Lung Opacity, L2: Pleural Effusion, L3: Atelectasis, L4: Enlarged Cardiac Silhouette, L5: Pulmonary Edema/Hazy Opacity, L6: Pneumothorax, L7: Consolidation, L8: Fluid Overload/Heart Failure, L9: Pneumonia.

| Method | L1 | L2 | L3 | L4 | L5 | L6 | L7 | L8 | L9 | **AVG** |
|---|---|---|---|---|---|---|---|---|---|---|
| Faster R-CNN | 0.84 | 0.89 | 0.77 | 0.85 | 0.87 | 0.77 | 0.75 | 0.81 | 0.71 | 0.80 |
| GlobalView | **0.91** | **0.94** | 0.86 | 0.92 | 0.92 | **0.93** | 0.86 | 0.87 | 0.84 | 0.89 |
| CheXGCN | 0.86 | 0.90 | **0.91** | **0.94** | **0.95** | 0.75 | **0.89** | **0.98** | **0.88** | **0.90** |

**Example Task 1: Change between sequential CXR exams.** CXRs are commonly repeatedly requested in the clinical workflow to assess for a myriad of attributes. Given a patient with sequential CXRs, the goal of this task is to automatically evaluate disease change over time based on two sequential CXR exams. We restricted the problem to a subset of the Chest ImaGenome dataset, i.e., to attributes related to congestive heart failure (CHF), as fluid management is one of the most routine clinical tasks for which CXRs can be ordered to guide the next steps (e.g. whether to give more intravenous fluid or give diuretics, etc). However, we note that users of this dataset can also explore comparison changes for other CXR attributes (e.g. pneumonia). Each CXR image is also associated with a bounding box that marks a localized area, e.g., "left lung" for specific anatomical finding (i.e., attribute), such as "pulmonary edema/hazy opacity", etc. In addition, the pair of CXR images is mapped to the comparison label that indicates whether the condition of the anatomical finding has improved or worsened. As a baseline example, we focus on change relations in the 'left lung' and 'right lung' objects that are related to the 'pulmonary edema/hazy opacity' and 'fluid overload/heart failure' attributes. The number of examples labeled in the training, validation and test data are $10,515$, $1,493$ and $2,987$, respectively. We design a siamese architecture (Figure 10 in Supplementary F) that first extracts the localized bounding box from each image and encodes the extracted image patches with a pre-trained ResNet101 autoencoder, denoted that is trained on several medical imaging datasets, e.g., NIH, CheXpert, and MIMIC datasets, etc. [4, 1, 3]. The autoencoder image representations are concatenated and passed through a dense layer with 128 neurons and ReLU activations, and a final classification layer. We train for 300 epochs with cross-entropy, stochastic gradient descent, $1e-3$ learning rate, $0.1$ gradient clipping and 32 batch size. We freeze the autoencoder weights and finetune the two last dense layers. On this challenging task of predicting change in localized anatomical findings between two sequential exams, we achieve an accuracy of $75.3\%$.

**Example Task 2: Localization of CXR attributes.** Knowing the anatomical location of non-specific findings/attributes on CXR images can help with narrowing down possible disease diagnoses and guide the next steps in requesting more specific imaging exams or treatment. To this end, we train a Faster R-CNN model [47] to learn 18 anatomical locations within the dataset. We extract the 1024 dimension convolution feature vector of each anatomical region. We re-implement the state-of-the-art CheXGCN model [48] to learn the dependencies between attributes within the Chest X-ray. Similar to the work done by CheXGCN we model the correlation of the CXR attributes using a conditional probability (see Figure 11 in Supplementary F). We compare the results of the model with two baseline models, a Faster R-CNN model followed by a linear model without the GCN, and a Densenet model [49] without the Faster R-CNN to evaluate the effectiveness of the localized models. We focus on 9 common CXR attributes, which include lung opacity, pleural effusion, atelectasis, enlarged cardiac silhouette, pulmonary edema/hazy opacity, pneumothorax, consolidation, fluid overload/heart failure, pneumonia. The results of the experiments are shown in Table 5 and the labels are ordered according to the attribute list above.

**Dataset Limitations**: The Chest ImaGenome dataset came from only one U.S. hospital source. It is automatically generated and is limited by the performance of the NLP and the Bbox extraction pipelines. Furthermore, we cannot assume that all the clinically relevant CXR attributes are always described on every exam by the reporting radiologists. In fact, we have observed many implied object-attribute relation descriptions that are documented only in the form of comparisons (e.g. no change from previous) in short CXR reports. As such, even with perfect NLP extraction of object and attribute relations from individual reports, there would be missing information in the report knowledge graph constructed for some images. These technical areas are worth improving on in future research with more powerful NLP, image processing techniques and other graph-based techniques. Addressing missing relations will certainly improve this dataset too. Regardless, version 1.0.0 of the Chest ImaGenome dataset serves as a pioneering vision for a richer radiology imaging dataset.

## Acknowledgements

This work was supported by the Rensselaer-IBM AI Research Collaboration, part of the IBM AI Horizons Network, and the IBM-MIT Critical Data Collaboration.

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

 # A Additional Chest ImaGenome Terminology Descriptions

Table 6: Semantic category of nodes and edges in CXR knowledge graphs. All nodes are mapped to UMLS CUIs in the scene graph jsons. All object nodes have corresponding bounding box coordinates on frontal CXRs except ones with *. All nodes and edges are evaluated with the gold standard dataset except the edges marked with **, which are modifiers of the context edges.

| Category ID | type | names |
|---|---|---|
| technicalassessment | attribute node | low lung volumes, rotated, artifact, breast/nipple shadows, skin fold |
| texture | attribute node | opacity, alveolar, interstitial, calcified, lucency |
| anatomicalfinding | attribute node | lung opacity, airspace opacity, consolidation, infiltration, atelectasis, linear/patchy atelectasis, lobar/segmental collapse, pulmonary edema/hazy opacity, vascular congestion, vascular redistribution, increased reticular markings/ild pattern, pleural effusion, costophrenic angle blunting, pleural/parenchymal scarring, bronchiectasis, enlarged cardiac silhouette, mediastinal displacement, mediastinal widening, enlarged hilum, tortuous aorta, vascular calcification, pneumomediastinum, pneumothorax, hydropneumothorax, lung lesion, mass/nodule (not otherwise specified), multiple masses/nodules, calcified nodule, superior mediastinal mass/enlargement, rib fracture, clavicle fracture, spinal fracture, hyperaeration, cyst/bullae, elevated hemidiaphragm, diaphragmatic eventration (benign), subdiaphragmatic air, subcutaneous air, hernia, scoliosis, spinal degenerative changes, shoulder osteoarthritis, bone lesion |
| disease | attribute node | pneumonia, fluid overload/heart failure, copd/emphysema, granulomatous disease, interstitial lung disease, goiter, lung cancer, aspiration, alveolar hemorrhage, pericardial effusion |
| nlp | attribute node | abnormal, normal (with respect to an anatomy/object node) |
| tubesandlines | attribute node | chest tube, mediastinal drain, pigtail catheter, endotracheal tube, tracheostomy tube, picc, ij line, chest port, subclavian line, swan-ganz catheter, intra-aortic balloon pump, enteric tube |
| device | attribute node | sternotomy wires, cabg grafts, aortic graft/repair, prosthetic valve, cardiac pacer and wires |
| majorstructure | object node | right lung, left lung, mediastinum |
| subanatomy | object node | right apical zone, right upper lung zone, right mid lung zone, right lower lung zone, right hilar structures, right costophrenic angle, left apical zone, left upper lung zone, left mid lung zone, left lower lung zone, left hilar structures, left costophrenic angle, upper mediastinum, cardiac silhouette, trachea, right hemidiaphragm, left hemidiaphragm, right clavicle, left clavicle, spine, right atrium, cavoatrial junction, svc, carina, aortic arch, abdomen, right chest wall*, left chest wall*, right shoulder*, left shoulder*, neck*, right arm*, left arm*, right breast*, left breast* |
| context | edge | yes (has/present in), no (not have/not present in) |
| comparison | edge | improved, worsened, no change |
| severity** | edge | hedge, mild, moderate, severe |
| temporal** | edge | acute, chronic |

## B  Scene Graph JSON

Below are examples from a scene graph JSON used for explanation for the silver dataset.

### B.1  Scene Graph JSON - first level

```
{
 'chest_imageimage_id': '10cd06e9-5443fef9-9afbe903-e2ce1eb5-dcff1097',
 'viewpoint': 'AP', 'patient_id': 10063856, 'study_id': 56759094,
 'gender': 'F', 'age_decile': '50-60',
 'reason_for_exam': '___F with hypotension.  Evaluate for pneumonia.',
 'StudyOrder': 2, 'StudyDateTime': '2178-10-05 15:05:32 UTC',
 'objects': [ <...list of {} for each object...> ],
 'attributes':[ <...list of {} for each object...> ],
 'relationships':[ <...list of {} of comparison relationships between objects
 from sequential exams for the same patient...> ]
}
```

### B.2  Scene Graph JSON - objects field

```
{
 'object_id': '10cd06e9-5443fef9-9afbe903-e2ce1eb5-dcff1097_right upper lung zone',
 'x1': 48, 'y1': 39, 'x2': 111, 'y2': 93,
 'width': 63, 'height': 54,
 'bbox_name': 'right upper lung zone',
 'synsets': ['C0934570'],
 'name': 'Right upper lung zone',
 'original_x1': 395, 'original_y1': 532,
 'original_x2': 1255, 'original_y2': 1268,
 'original_width': 860, 'original_height': 736
}
```

### B.3  Scene Graph JSON - attributes field

```
{
 'right lung': True, 'bbox_name': 'right lung',
 'synsets': ['C0225706'], 'name': 'Right lung',
 'attributes': [['anatomicalfinding|no|lung opacity',
 'anatomicalfinding|no|pneumothorax',  'nlp|yes|normal'],
 ['anatomicalfinding|no|pneumothorax']],
 'attributes_ids': [['CL556823', 'C1963215;;C0032326', 'C1550457'],
 ['C1963215;;C0032326']],
 'phrases': ['Right lung is clear without pneumothorax.',
 'No pneumothorax identified.'],
 'phrase_IDs': ['56759094|10', '56759094|14'],
 'sections': ['finalreport', 'finalreport'],
 'comparison_cues': [[], []],
 'temporal_cues': [[], []],
 'severity_cues': [[], []],
 'texture_cues': [[], []],
 'object_id': '10cd06e9-5443fef9-9afbe903-e2ce1eb5-dcff1097_right lung'
}
```

### B.4  Scene Graph JSON - relationships field

```
{
 'relationship_id': '56759094|7_54814005_C0929215_10cd06e9_4bb710ab',
 'predicate': ''['No status change']'',
 'synsets': ['C0442739'],
 'relationship_names': ['comparison|yes|no change'],
 'relationship_contexts': [1.0],
 'phrase': 'Compared with the prior radiograph, there is a persistent veil
 -like opacity\n over the left hemithorax, with a crescent of air surrounding
```

```
720    the aortic arch,\n in keeping with continued left upper lobe collapse.',
721    'attributes': ['anatomicalfinding|yes|atelectasis',
722    'anatomicalfinding|yes|lobar/segmental collapse',
723    'anatomicalfinding|yes|lung opacity', 'nlp|yes|abnormal'],
724    'bbox_name': 'left upper lung zone',
725    'subject_id': '10cd06e9-5443fef9-9afbe903-e2ce1eb5-dcff1097_left upper lung zone',
726    'object_id': '4bb710ab-ab7d4781-568bcd6e-5079d3e6-7fdb61b6_left upper lung zone'
727  }
```

## B.5  Scene Graph - Enriched RDF JSON format

```
729  {
730   <study_id_i> : [
731                    [[node_id_1, node_type_1], [node_id_2, node_type_2], relation_name_A],
732                    [[node_id_1, node_type_1], [node_id_3, node_type_3], relation_name_B],
733                        ...
734               ],
735   <study_id_i+1>:[
736                    [[node_id_1, node_type_1], [node_id_2, node_type_2], relation_name_A],
737                    [[node_id_1, node_type_1], [node_id_3, node_type_3], relation_name_B],
738                        ...
739               ],
740  }
```

# C  Gold Dataset Annotation - Details

The 'gold dataset' is a randomly sampled subset (500 unique patients) from the automatically generated Chest ImaGenome dataset, i.e., the 'silver dataset', that has been manually validated or corrected. The primary purpose of the 'gold dataset' is to evaluate the quality of labels in the 'silver dataset'. For this purpose, we evaluated the Chest ImaGenome dataset along with the 3 components below (A-B). The annotations for each component were collected in stages to reduce the cognitive workload for the annotators. The annotators are all M.D.s with 2 to 10 or more years of clinical experience. One of the annotators is a radiologist trained in the United States, who has over 6 years of radiology experience and specializes in reading imaging exams from the Emergency Department (ED) setting. The annotation tasks were delegated to the annotators according to their clinical experience, which we think are all more than sufficient for the tasks. Component A and B were annotated by the radiologist and an M.D. and component C was annotated by 4 M.D.'s.

*A) Evaluating CXR knowledge graph extraction from reports*

The report knowledge graph for the *first* CXR of the 500 patients was manually reviewed and corrected as necessary for relation extraction between the anatomical locations (objects) and the CXR attributes. From piloting trials, we found that manually annotating multiple targets at a document level lead to a slow and complex task with poor recall. However, sometimes information from prior sentences is necessary to annotate both the anatomical locations and the attributes correctly. Therefore, we set up the annotation task at the sentence level. Sentences from each report are ordered as per the original report, and the phrase boundary for each attribute was marked out for the annotators, where the phrases used for detecting each attribute were curated by consensus between two radiologists from previous work [5].

Since we are targeting a large set of possible anatomical locations (object) to attribute combinations, the annotation was streamlined into the four steps below to minimize the cognitive overload for each step. Steps 1 and 2 are dual annotated by two clinicians (one fully trained radiologist and one M.D.), with disagreements resolved by consensus review. Steps 3 and 4 are single annotated. A random subset of annotations for 500 sentences from step 4 are sampled and dual annotated to estimate inter-annotator agreement. Cleaned results from step 4 constitute the final gold-standard CXR knowledge graph ground truth for the 500 reports.

This annotation component was set up in Excel and was broken down into the following four steps below. In our Excel setup, all sentences from each report are available to the annotators (they can just

scroll up or down). The sentences are ordered by 'row_id' sequentially within each report. Unique patients and reports have the same IDs as shown in the figures below.

**Step 1** - For each sentence and NLP extracted attribute combination, decide whether the NLP context (affirmed or negated) for the attribute was correct. If not, correct it. Figure 3 shows how this task was set up in Excel. The annotators' task is to make sure the extracted attribute (yellow label_name column) has the correct context given the sentence from the report. This 'context' is used as the relation between the location and the attribute in the final annotated result.

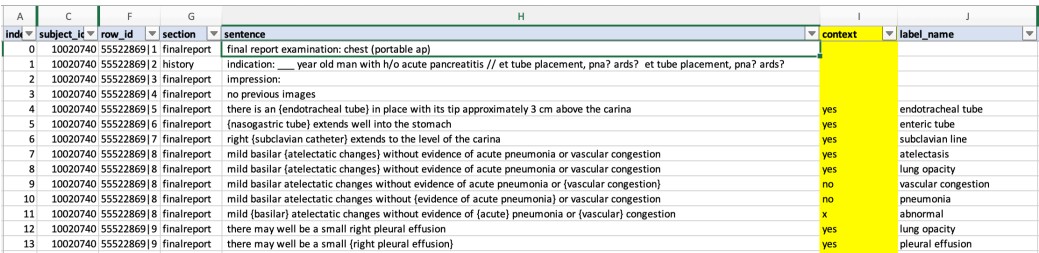

Figure 3: Step 1: Annotate all attributes per sentence.

**Step 2** - For each sentence, decide whether the NLP extracted anatomical location(s) were described or implied by the reporting radiologist. If not, remove the location (in yellow column 'bboxes_corrected). If missing, add the location. If unsure (e.g., if lung is mentioned but not sure if it is the right or left lung), the annotator can look in previous sentences from the same report. The task was set up as shown in Figure 4.

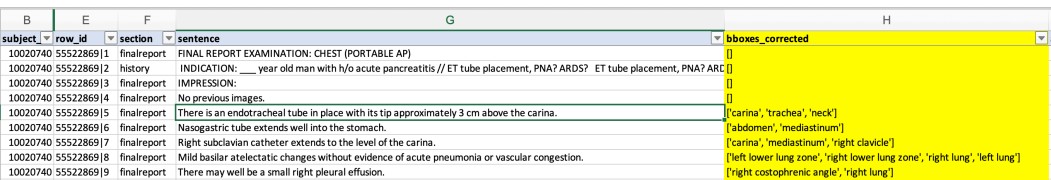

Figure 4: Step 2: Annotate all locations per sentence.

**Step 3** - For recall, manually annotate missed objects and/or attributes for sentences with no NLP extractions (a much smaller subset). For this, we used Excel's filtering function to look at all sentences with no automated extractions (empty cells) and de novo added the manual annotations.

**Step 4** - Firstly, all rows from steps 1-3 where the annotations differed between the two annotators were reviewed and resolved together by consensus. Then we automatically derived all object-attribute relation combinations for each sentence from steps 1-3's results. The obviously wrong object-to-attribute relations were filtered out for each sentence using the CXR ontology. For the remaining object-to-attribute relations for each sentence, the task was to indicate whether the logical statement of *"object X contains (or does not contain) attribute Y"* is true or false, as shown in Figure 5. Probable relation is still defined to be true for this annotation. Annotating for uncertain relations is beyond the scope of this project. However, for future dataset expansion, we have kept the NLP cues for the certainty for each object-attribute relation in the scene graph JSON.

Since step 4 was single annotated, to estimate the final inter-annotator agreement, we randomly sampled 500 sentences for dual annotations. This annotated result is also shared on PhysioNet.

*B) Evaluating comparison relation extraction*:

The *second* CXR exam report for the 500 patients was reviewed for comparison relation extraction. The annotation was also set up in Excel and conducted at the sentence level. However, the annotator is also shown the whole previous CXR report for context. Similarly, we split the annotation task up into several steps, where steps 1 and 2 are dual annotated and disagreement resolved via consensus. Steps 3 and 4 were single annotated. A subset of 500 sentences from the final annotations was reviewed by a second annotator for assessing inter-annotator agreement.

| patient_id | row_id | section | bbox | relation | label_name | sentence |
|---|---|---|---|---|---|---|
| 10020740 | 55522869\|5 | finalreport | trachea | 1 | endotracheal tube | There is an endotracheal tube in place with its tip approximately 3 cm above the carina. |
| 10020740 | 55522869\|6 | finalreport | abdomen | 1 | enteric tube | Nasogastric tube extends well into the stomach. |
| 10020740 | 55522869\|6 | finalreport | mediastinum | 1 | enteric tube | Nasogastric tube extends well into the stomach. |
| 10020740 | 55522869\|6 | finalreport | neck | 1 | enteric tube | Nasogastric tube extends well into the stomach. |
| 10020740 | 55522869\|7 | finalreport | mediastinum | 1 | subclavian line | Right subclavian catheter extends to the level of the carina. |
| 10020740 | 55522869\|7 | finalreport | right clavicle | 1 | subclavian line | Right subclavian catheter extends to the level of the carina. |
| 10020740 | 55522869\|8 | finalreport | left hilar structures | 0 | vascular congestion | Mild basilar atelectatic changes without evidence of acute pneumonia or vascular congestion. |
| 10020740 | 55522869\|8 | finalreport | left lower lung zone | 0 | pneumonia | Mild basilar atelectatic changes without evidence of acute pneumonia or vascular congestion. |
| 10020740 | 55522869\|8 | finalreport | left lower lung zone | 0 | vascular congestion | Mild basilar atelectatic changes without evidence of acute pneumonia or vascular congestion. |
| 10020740 | 55522869\|8 | finalreport | left lower lung zone | 1 | abnormal | mild basilar {atelectatic changes} without evidence of acute pneumonia or vascular congestion |
| 10020740 | 55522869\|8 | finalreport | left lower lung zone | 1 | atelectasis | Mild basilar atelectatic changes without evidence of acute pneumonia or vascular congestion. |
| 10020740 | 55522869\|8 | finalreport | left lower lung zone | 1 | lung opacity | Mild basilar atelectatic changes without evidence of acute pneumonia or vascular congestion. |
| 10020740 | 55522869\|8 | finalreport | left lung | 0 | pneumonia | Mild basilar atelectatic changes without evidence of acute pneumonia or vascular congestion. |
| 10020740 | 55522869\|8 | finalreport | left lung | 0 | vascular congestion | Mild basilar atelectatic changes without evidence of acute pneumonia or vascular congestion. |
| 10020740 | 55522869\|8 | finalreport | left lung | 1 | abnormal | mild basilar {atelectatic changes} without evidence of acute pneumonia or vascular congestion |
| 10020740 | 55522869\|8 | finalreport | left lung | 1 | atelectasis | Mild basilar atelectatic changes without evidence of acute pneumonia or vascular congestion. |
| 10020740 | 55522869\|8 | finalreport | left lung | 1 | lung opacity | Mild basilar atelectatic changes without evidence of acute pneumonia or vascular congestion. |
| 10020740 | 55522869\|8 | finalreport | right hilar structures | 0 | vascular congestion | Mild basilar atelectatic changes without evidence of acute pneumonia or vascular congestion. |
| 10020740 | 55522869\|8 | finalreport | right lower lung zone | 0 | pneumonia | Mild basilar atelectatic changes without evidence of acute pneumonia or vascular congestion. |
| 10020740 | 55522869\|8 | finalreport | right lower lung zone | 0 | vascular congestion | Mild basilar atelectatic changes without evidence of acute pneumonia or vascular congestion. |
| 10020740 | 55522869\|8 | finalreport | right lower lung zone | 1 | abnormal | mild basilar {atelectatic changes} without evidence of acute pneumonia or vascular congestion |
| 10020740 | 55522869\|8 | finalreport | right lower lung zone | 1 | atelectasis | Mild basilar atelectatic changes without evidence of acute pneumonia or vascular congestion. |
| 10020740 | 55522869\|8 | finalreport | right lower lung zone | 1 | lung opacity | Mild basilar atelectatic changes without evidence of acute pneumonia or vascular congestion. |
| 10020740 | 55522869\|8 | finalreport | right lung | 0 | pneumonia | Mild basilar atelectatic changes without evidence of acute pneumonia or vascular congestion. |
| 10020740 | 55522869\|8 | finalreport | right lung | 0 | vascular congestion | Mild basilar atelectatic changes without evidence of acute pneumonia or vascular congestion. |
| 10020740 | 55522869\|8 | finalreport | right lung | 1 | abnormal | mild basilar {atelectatic changes} without evidence of acute pneumonia or vascular congestion |
| 10020740 | 55522869\|8 | finalreport | right lung | 1 | atelectasis | Mild basilar atelectatic changes without evidence of acute pneumonia or vascular congestion. |

Figure 5: Step 4: Annotate all logically correct statements/relations for each sentence.

**Step 1** - Given the previous report and the current report sentence, decide whether the extracted comparison cue(s) (improved, worsened, no change) is/are correct. If not, correct it/them. In this step, the annotators are asked to validate or correct the column 'comparison' in Figure 6.

**Step 2** - Building from step 1 for each sentence, given a validated or corrected comparison cue, validate whether all the anatomical location(s) extracted are correct (column 'bbox' in Figure 6). If incorrect or missing, remove or add the correct location(s) to the column.

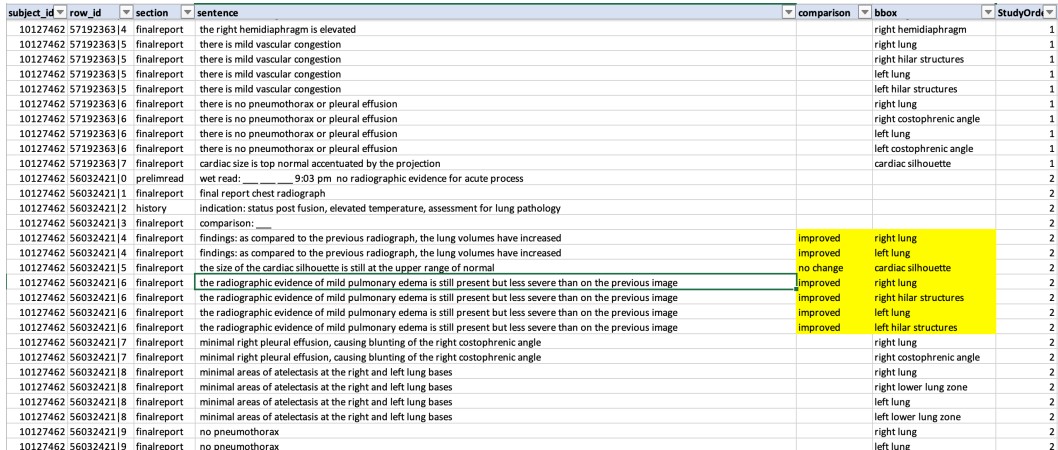

| subject_id | row_id | section | sentence | comparison | bbox | StudyOrder |
|---|---|---|---|---|---|---|
| 10127462 | 57192363\|4 | finalreport | the right hemidiaphragm is elevated | | right hemidiaphragm | 1 |
| 10127462 | 57192363\|5 | finalreport | there is mild vascular congestion | | right lung | 1 |
| 10127462 | 57192363\|5 | finalreport | there is mild vascular congestion | | right hilar structures | 1 |
| 10127462 | 57192363\|5 | finalreport | there is mild vascular congestion | | left lung | 1 |
| 10127462 | 57192363\|5 | finalreport | there is mild vascular congestion | | left hilar structures | 1 |
| 10127462 | 57192363\|6 | finalreport | there is no pneumothorax or pleural effusion | | right lung | 1 |
| 10127462 | 57192363\|6 | finalreport | there is no pneumothorax or pleural effusion | | right costophrenic angle | 1 |
| 10127462 | 57192363\|6 | finalreport | there is no pneumothorax or pleural effusion | | left lung | 1 |
| 10127462 | 57192363\|6 | finalreport | there is no pneumothorax or pleural effusion | | left costophrenic angle | 1 |
| 10127462 | 57192363\|7 | finalreport | cardiac size is top normal accentuated by the projection | | cardiac silhouette | 1 |
| 10127462 | 56032421\|0 | prelimread | wet read: ___ ___ 9:03 pm  no radiographic evidence for acute process | | | 2 |
| 10127462 | 56032421\|1 | finalreport | final report chest radiograph | | | 2 |
| 10127462 | 56032421\|2 | history | indication: status post fusion, elevated temperature, assessment for lung pathology | | | 2 |
| 10127462 | 56032421\|3 | finalreport | comparison: ___ | | | 2 |
| 10127462 | 56032421\|4 | finalreport | findings: as compared to the previous radiograph, the lung volumes have increased | improved | right lung | 2 |
| 10127462 | 56032421\|4 | finalreport | findings: as compared to the previous radiograph, the lung volumes have increased | improved | left lung | 2 |
| 10127462 | 56032421\|5 | finalreport | the size of the cardiac silhouette is still at the upper range of normal | no change | cardiac silhouette | 2 |
| 10127462 | 56032421\|6 | finalreport | the radiographic evidence of mild pulmonary edema is still present but less severe than on the previous image | improved | right lung | 2 |
| 10127462 | 56032421\|6 | finalreport | the radiographic evidence of mild pulmonary edema is still present but less severe than on the previous image | improved | right hilar structures | 2 |
| 10127462 | 56032421\|6 | finalreport | the radiographic evidence of mild pulmonary edema is still present but less severe than on the previous image | improved | left lung | 2 |
| 10127462 | 56032421\|6 | finalreport | the radiographic evidence of mild pulmonary edema is still present but less severe than on the previous image | improved | left hilar structures | 2 |
| 10127462 | 56032421\|7 | finalreport | minimal right pleural effusion, causing blunting of the right costophrenic angle | | right lung | 2 |
| 10127462 | 56032421\|7 | finalreport | minimal right pleural effusion, causing blunting of the right costophrenic angle | | right costophrenic angle | 2 |
| 10127462 | 56032421\|8 | finalreport | minimal areas of atelectasis at the right and left lung bases | | right lung | 2 |
| 10127462 | 56032421\|8 | finalreport | minimal areas of atelectasis at the right and left lung bases | | right lower lung zone | 2 |
| 10127462 | 56032421\|8 | finalreport | minimal areas of atelectasis at the right and left lung bases | | left lung | 2 |
| 10127462 | 56032421\|8 | finalreport | minimal areas of atelectasis at the right and left lung bases | | left lower lung zone | 2 |
| 10127462 | 56032421\|9 | finalreport | no pneumothorax | | right lung | 2 |
| 10127462 | 56032421\|9 | finalreport | no pneumothorax | | left lung | 2 |

Figure 6: Step 1 and 2: Annotate change relations for different anatomical locations

**Step 3** - Building from step 2 for each sentence, given each correct comparison cue and anatomical location relation, decide whether the attributes assigned to the location described or implied in the sentence are correct or not. If not, correct it. Figure 7 illustrates how step 3 was set up, where the annotators' task is to validate or correct the 'label_name' column with respect to the 'bbox', 'relation' and 'comparison' columns for each sentence.

**Step 4** - For recall, we used the filtering function in Excel to isolate all sentences with no comparison cue extractions from step 3. Sentences with missing comparison annotations were manually de-novo annotated.

*C) Evaluating anatomy object detection for CXR images*:

The first and second CXR images for the same 500 patients were dual validated and corrected for the bounding box objects (i.e., 1000 frontal CXR images altogether). Given the resources we had,

| patient_id | study_id | studyOrd | row_id | section | bbox | relation | label_name | comparison | sentence |
|---|---|---|---|---|---|---|---|---|---|
| 10127462 | 56032421 | 2 | 56032421\|4 | finalreport | left lung | | 1 low lung volumes | ['improved'] | findings: as compared to the previous radiograph, the lung volumes have increased |
| 10127462 | 56032421 | 2 | 56032421\|4 | finalreport | right lung | | 1 low lung volumes | ['improved'] | findings: as compared to the previous radiograph, the lung volumes have increased |
| 10127462 | 56032421 | 2 | 56032421\|5 | finalreport | cardiac silhouette | | 0 enlarged cardiac silhouette | ['no change'] | the size of the cardiac silhouette is still at the upper range of normal |
| 10127462 | 56032421 | 2 | 56032421\|5 | finalreport | cardiac silhouette | | 1 normal | ['no change'] | the size of the cardiac silhouette is still at the upper range of normal |
| 10127462 | 56032421 | 2 | 56032421\|6 | finalreport | left hilar structures | | 1 abnormal | ['improved'] | the radiographic evidence of mild pulmonary edema is still present but less severe than on the previous image |
| 10127462 | 56032421 | 2 | 56032421\|6 | finalreport | left hilar structures | | 1 lung opacity | ['improved'] | the radiographic evidence of mild pulmonary edema is still present but less severe than on the previous image |
| 10127462 | 56032421 | 2 | 56032421\|6 | finalreport | left hilar structures | | 1 pulmonary edema/hazy opacity | ['improved'] | the radiographic evidence of mild pulmonary edema is still present but less severe than on the previous image |
| 10127462 | 56032421 | 2 | 56032421\|6 | finalreport | left lung | | 1 abnormal | ['improved'] | the radiographic evidence of mild pulmonary edema is still present but less severe than on the previous image |
| 10127462 | 56032421 | 2 | 56032421\|6 | finalreport | left lung | | 1 lung opacity | ['improved'] | the radiographic evidence of mild pulmonary edema is still present but less severe than on the previous image |
| 10127462 | 56032421 | 2 | 56032421\|6 | finalreport | left lung | | 1 pulmonary edema/hazy opacity | ['improved'] | the radiographic evidence of mild pulmonary edema is still present but less severe than on the previous image |
| 10127462 | 56032421 | 2 | 56032421\|6 | finalreport | right hilar structures | | 1 abnormal | ['improved'] | the radiographic evidence of mild pulmonary edema is still present but less severe than on the previous image |
| 10127462 | 56032421 | 2 | 56032421\|6 | finalreport | right hilar structures | | 1 lung opacity | ['improved'] | the radiographic evidence of mild pulmonary edema is still present but less severe than on the previous image |
| 10127462 | 56032421 | 2 | 56032421\|6 | finalreport | right hilar structures | | 1 pulmonary edema/hazy opacity | ['improved'] | the radiographic evidence of mild pulmonary edema is still present but less severe than on the previous image |
| 10127462 | 56032421 | 2 | 56032421\|6 | finalreport | right lung | | 1 abnormal | ['improved'] | the radiographic evidence of mild pulmonary edema is still present but less severe than on the previous image |
| 10127462 | 56032421 | 2 | 56032421\|6 | finalreport | right lung | | 1 lung opacity | ['improved'] | the radiographic evidence of mild pulmonary edema is still present but less severe than on the previous image |
| 10127462 | 56032421 | 2 | 56032421\|6 | finalreport | right lung | | 1 pulmonary edema/hazy opacity | ['improved'] | the radiographic evidence of mild pulmonary edema is still present but less severe than on the previous image |

Figure 7: Step 3: Annotate change relations for different anatomical locations with respect to attribute

we selected 28 anatomical objects (out of 36 available) that are clinically most important for frontal CXRs interpretations. The automatically extracted bounding box coordinates were first plotted on resized and padded 224x224 images. From piloting, we determined that this image size is sufficiently large to annotate the anatomies that we were targeting. The plotted images were displayed one at a time to annotators via a custom Jupyter Notebook that we had set up to allow bounding box coordinates and label annotations. We set up the annotation task on two panels, one for lung-related bounding boxes (Figure 8) and another for mediastinum related and other bounding boxes (Figure 9).

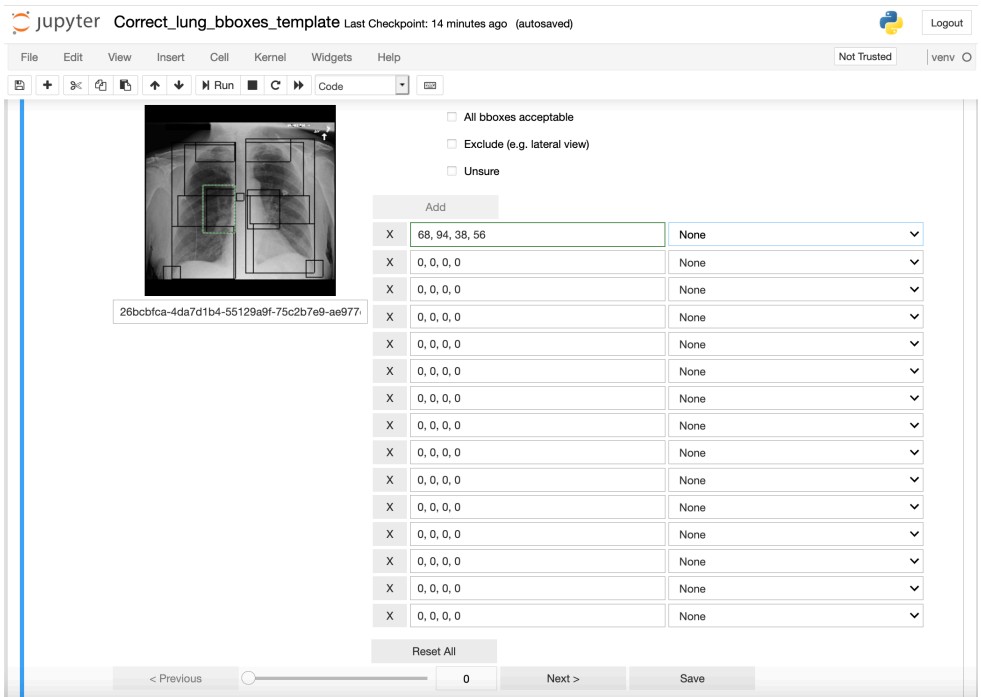

Figure 8: Bbox annotations - lung related Bboxes panel

Four M.D.'s were trained to perform this task after reviewing a set of 20-30 training examples with a radiologist. Since the inter-annotator agreement is high (mean IoU > 0.96 for all objects), the final cleaned gold standard bbox coordinates use the average coordinates from two annotators for each bounding box.

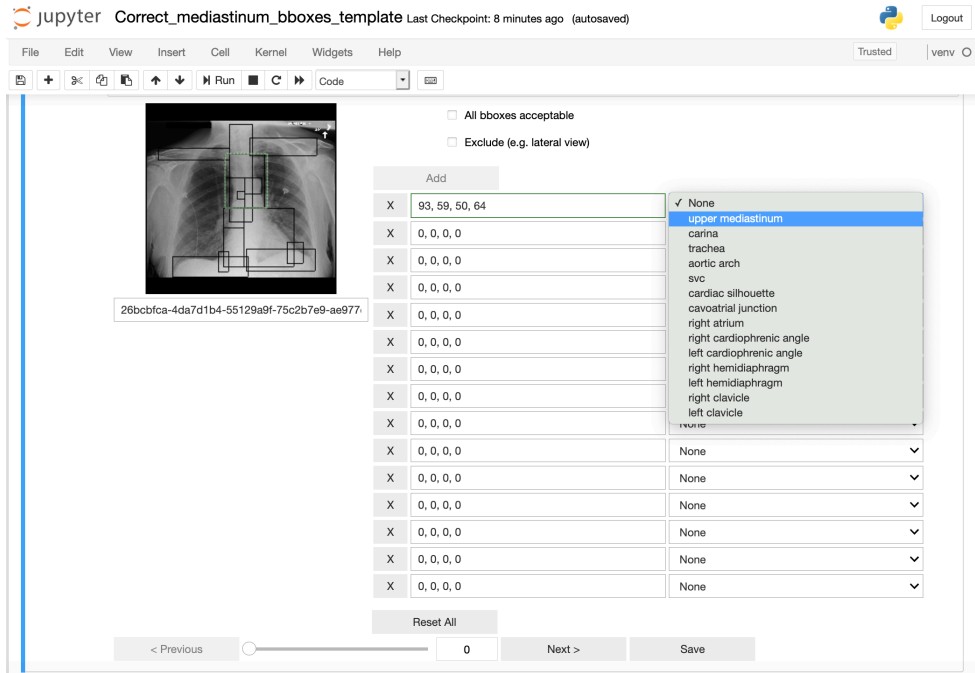

Figure 9: Bbox annotations - mediastinum related and other Bboxes panel

## D  Dataset Usage Supporting Files

**gold_all_sentences_500pts_1000studies.txt** contains all the sentences tokenized from the original MIMIC-CXR reports that were used to create the gold standard dataset. We include this file because sentences with no relevant object, attribute or relation descriptions did not make it into the gold standard dataset. We renamed 'subject_id' from MIMIC-CXR dataset to 'patient_id' in Chest ImaGenome dataset to avoid confusion with field names for relationships in the scene graphs. Otherwise, the ids are unchanged. Sentences in the tokenized file are assigned to 'history', 'prelimread', or 'finalreport' in the 'section' column. The 'sent_loc' column contains the order of the sentences as in the original report. Minimal tokenization has been done to the sentences.

**gold_bbox_scaling_factors_original_to_224x224.csv** contains the scaling 'ratio' and the paddings ('left', 'right', 'top', and 'bottom') added to square the image after resizing the original MIMIC-CXR dicoms to 224x224 sizes. These ratios were used to rescale the annotated coordinates for 224x224 images back to the original CXR image sizes.

**auto_bbox_pipeline_coordinates_1000_images.txt** contains the bounding box coordinates that were automatically extracted by the Bbox pipeline for the different objects for images in the gold standard dataset. It is in a tabular format like with the ground truth for easier evaluation purposes.

**object-bbox-coordinates_evaluation.ipynb** notebook calculates the bounding box object detection performance using ground truth files from the 4 M.D. annotators , as well as consolidating the final **gold_bbox_coordinate_annotations_1000images.csv**.

**Preprocess_mimic_cxr_v2.0.0_reports.ipynb** processes the reports (tokenize sentences and sort them into history, prelim or final report sentences) from the original MIMIC-CXR v2.0.0 and save output as **silver_dataset/cxr-mimic-v2.0.0-processed-sentences_all.txt**. Only sentences with object or attribute extractions ended up in the final scene graph jsons in the Chest ImaGenome dataset.

The **semantics** directory contains the object (**objects_detectable_by_bbox_pipeline_v1.txt** and **objects_extracted_from_reports_v1.txt**), attribute (**attribute_relations_v1.txt**) and comoparison (**comparison_relations_v1.txt**) relations labels in the Chest ImaGenome dataset. It also contains **semantics/label_to_UMLS_mapping.json**, which maps all Chest ImaGenome concepts to UMLS CUIs [34].

# E   Dataset Evaluation

Table 7 reports anatomical location level object-to-attribute relations extraction performance by the scene graph extraction pipeline. The report numbers are calculated by a combination of notebooks: 'generate_scenegraph_statistics.ipynb', 'object-attribute-relation_evaluation.ipynb' and 'object-bbox-coordinates_evaluation.ipynb'.

Table 7: CXR image object detection evaluation results. * These anatomical locations are extracted by the Bbox pipeline but they are not manually annotated in the gold standard dataset due to resource constraints. ** The mediastinum bounding boxes were not directly annotated due to resource constraints. Mediastinum's bounding box boundary can be derived from the ground truth for the upper mediastinum and the cardiac silhouette.

| Bbox name (object) | Object-attribute relations frequency (500 reports) | Relationships F1 (500 reports) | Bbox IoU (over 1000 images) | % Bboxes corrected (1000 images) | % Relations missing Bbox coordinates (over whole dataset) |
|---|---|---|---|---|---|
| left lung | 1453 | 0.933 | 0.976 | 9.90% | 0.03% |
| right lung | 1436 | 0.937 | 0.983 | 6.30% | 0.04% |
| cardiac silhouette | 633 | 0.966 | 0.967 | 9.70% | 0.01% |
| mediastinum | 601 | 0.952 | ** | ** | 0.02% |
| left lower lung zone | 609 | 0.932 | 0.955 | 8.60% | 2.36% |
| right lower lung zone | 580 | 0.902 | 0.968 | 6.00% | 2.27% |
| right hilar structures | 572 | 0.934 | 0.976 | 4.10% | 1.91% |
| left hilar structures | 571 | 0.944 | 0.971 | 4.30% | 2.28% |
| upper mediastinum | 359 | 0.940 | 0.994 | 1.40% | 0.12% |
| left costophrenic angle | 298 | 0.908 | 0.929 | 9.60% | 0.63% |
| right costophrenic angle | 286 | 0.918 | 0.944 | 6.90% | 0.39% |
| left mid lung zone | 173 | 0.940 | 0.967 | 5.70% | 2.79% |
| right mid lung zone | 169 | 0.830 | 0.968 | 5.30% | 2.31% |
| aortic arch | 144 | 0.965 | 0.991 | 1.40% | 0.62% |
| right upper lung zone | 117 | 0.873 | 0.972 | 5.80% | 0.04% |
| left upper lung zone | 83 | 0.811 | 0.968 | 6.40% | 0.22% |
| right hemidiaphragm | 78 | 0.947 | 0.955 | 7.90% | 0.15% |
| right clavicle | 71 | 0.615 | 0.986 | 2.80% | 0.50% |
| left clavicle | 67 | 0.642 | 0.983 | 3.00% | 0.51% |
| left hemidiaphragm | 65 | 0.930 | 0.944 | 11.30% | 0.14% |
| right apical zone | 58 | 0.852 | 0.969 | 5.40% | 1.99% |
| trachea | 57 | 0.983 | 0.995 | 0.90% | 0.24% |
| left apical zone | 47 | 0.938 | 0.963 | 6.20% | 2.40% |
| carina | 41 | 0.975 | 0.994 | 0.80% | 1.47% |
| svc | 19 | 0.973 | 0.995 | 0.70% | 0.66% |
| right atrium | 14 | 0.963 | 0.979 | 4.00% | 0.18% |
| cavoatrial junction | 5 | 1.000 | 0.977 | 4.30% | 0.25% |
| abdomen | 80 | 0.904 | * | * | 0.26% |
| spine | 132 | 0.824 | * | * | 0.10% |

## F Pictorial Overview of Model Architectures

Due to space limitations, we present overview figures for the models designed for Example Tasks 1 and 2 here.

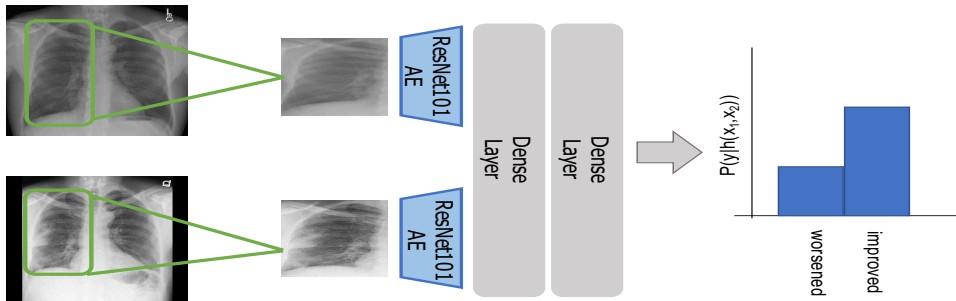

Figure 10: Example Task 1 Model Overview. Given a pair of CXR images, we extract features for the anatomical regions of interest with a pretrained ResNet autoencoder, concatenate representations and pass them through a dense layer and a final classification layer.

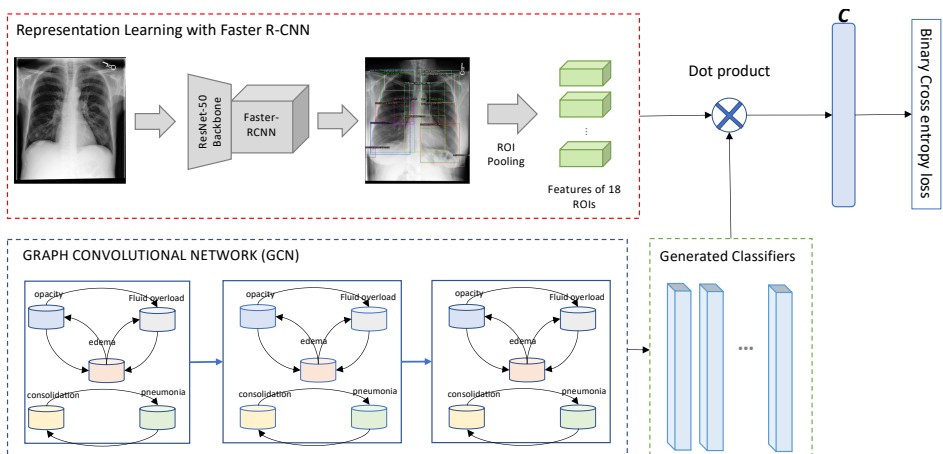

Figure 11: Example Task 2 Model Overview. Given a pair of CXR images, we extract features for the anatomical regions of interest with a pretrained Faster R-CNN and a GCN to learn the label dependencies.

## G    Qualitative Evaluation

In Figure 12, we visualize the output from our model for the anatomical finding predictions of costophrenic angles and enlarged cardiac silhouette. In Figure 13, we present an additional example, showing that the model is able to provide accurate localization information as well as predict the correct finding, i.e., showing accurate localization.

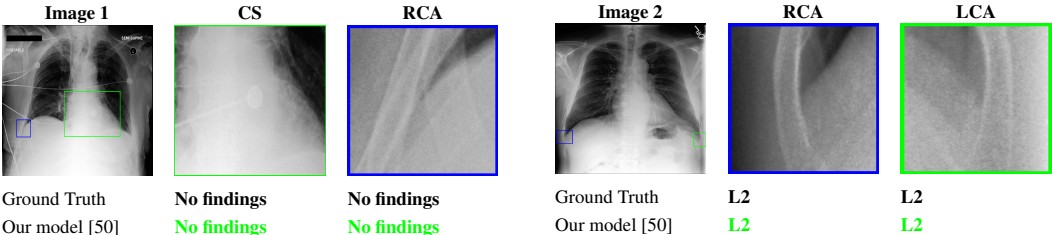

Figure 12: Examples of the prediction results. The overall chest X-ray image is shown alongside two anatomical regions, and predictions are compared against the ground-truth labels.

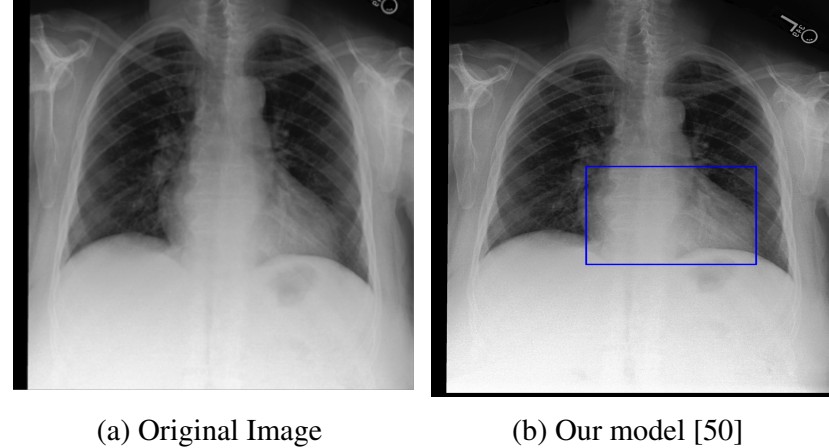

(a) Original Image        (b) Our model [50]

Figure 13: Example image with enlarged cardiac silhouette, showing that the trained model detects the finding in the correct bounding box.

