# OpenReview forum: "Chest ImaGenome Dataset for Clinical Reasoning"
_NeurIPS.cc/2021/Track/Datasets_and_Benchmarks/Round1 — Submitted to NeurIPS 2021 Datasets and Benchmarks Track (Round 1)_

### Official Review · Reviewer_xHyG · 2021-06-30

**Rating:** 6
**Confidence:** 3

**Strengths:**

This is the largest, to-date, dataset of CXR bounding box annotations and the first effort to expand the Visual Genome project to medical images. This efforts would be extremely valuable for driving medical image analysis using computational modeling, and addressing tasks such as classification, segmentation, and object detection.

**Weaknesses:**

My main concern is that it appears that the manuscript and presentation of the dataset is missing important details. I was not able to access the dataset via PsysioNet. The manuscript is missing qualitative visualization of the data and an analysis of how the additional data can improve performance of existing computational models. Given that the main contribution of the paper is the dataset, its ease of use and access should be top priority and should be re-evaluated.

 It also seems like the data already exists (MIMIC), but was additionally annotated and evaluated. The paper is missing details of how these annotators were selected (what are their credentials?), and what specific efforts were done over already published data and papers to create the current dataset.

**Additional Feedback:**

I think the paper has a great dataset to contribute, but needs to be more organized and clarified. The dataset an associated code also needs to be accessible. I would suggest that the authors work to address these points in order to improve the presentation.

**Clarity:**

The paper is a bit hard to follow. The method section and data description section can be significantly shortened to include the most important details, while other details can be moved to supplementary material. On the other hand, there are almost (except for teaser Figure 1) images with visualization of region annotations that would help explain the data.

I would also really like to see the Dataset evaluation section expanded, and observe how the data collected in the paper can improve performance compared to existing data.

Please include images with example annotated CXR regions in the manuscript to help explain the data.

What is silver dataset and how is it different from the Chest ImaGenome dataset? A consistent naming would be really helpful.

Line 86: “Table “ -> typo
Line 95: define AP and PA
Line 102: “Supplmentary” -> typo
Line 222: “Bbox pipeline” -> typo
Line 328: “21594” -> 21,594
Line 332: “5156” -> 5,156 Line 362: “1000” -> 1,000
Line 396: also commas (see above)


The supplementary file links to the paper and supplementary material. It would be more clear if it had just the supplementary material there.

Supplementary material:
Remove lines 582-589: these were examples lines and need to be removed.
Line 673 in the checklist: links to a broken DOI link

**Correctness:**

It appears that the dataset is a constructed in a sound way.


**Documentation:**


Data documentation, intended use, hosting, licencing and maintenance plans are included. I was not able to access the dataset via the PhysioNet website and see any of the described code, and thus cannot comment reproducibility.

I would suggest authors devote a few additional sentences to explaining how they use existing data to create and organize the proposed dataset. It appears that I cannot access the dataset as it is under review at PhysioNet currently, which limits my ability to evaluate the proposed data. However, it is well described in the text and

**Ethics:**

I am concerned that it seems like the authors are re-using existing data, but its not clear exactly how and whether any permissions should be sought for this. This process should be explained much more clearly in the submission.

**Relation To Prior Work:**

Yes, relation to prior work is clearly described in a table and in text.

**Summary And Contributions:**

The project collects a dataset of chest radiograph (CXR) anatomical locations, annotated via bounding boxes and scene graph data mined from MIMIC and PhysioNet. This is a valuable contribution as it allows for large-scale analysis of medical images where regions that are relevant to a specific diagnosis are labelled in the image.

---

> ### Author Response · Authors · 2021-07-14
> **Thank you for the comments. Please find our responses below**
>
> **Access**: Please note that the password for the review version of the dataset was supplied on OpenReview and was supposed to be passed on to reviewers. The dataset has been accepted on PhysioNet for public viewing (link https://doi.org/10.13026/wv01-y230), hence can be downloaded by any *MIMIC credentialed* researchers. Note that our dataset is in the form of a supplement to MIMIC-CXR and the original images need to be collected from the MIMIC-CXR PhysioNet repository.
>
> **Data already exists but was additionally annotated and evaluated:** Indeed, we significantly extend and add value to already available data (MIMIC). Note that the original dataset’s annotations were global (image-level) and restricted to just 14 CXR relevant labels, automatically extracted from CXR reports using the CheXpert NLP labeler. In previous work, nothing else about the image or the report has been structured in a way that reflects radiologists' reporting and reasoning processes. There are also no localized annotations from the original dataset release. Chest Imagenome adds much value to MIMIC-CXR by 1) providing localized annotations at the anatomical level, 2) describe medical imaging findings as an anatomically-centered graph, which is how radiologists are taught to read and reason through images, and 3) provides a framework to describe comparison relations between sequential medical imaging exams using anatomical units as points of comparison.
>
> **Annotators' details:**  Thank you for the valid point. All annotators are M.D.’s, one of whom is a seasoned American Board of Radiology certified radiologist (>6 years of radiology experience) who specializes in reading Emergency Department imaging exams. We were brief on the annotator credentials due to space limitations in our initial submission. We have revised supplementary material to include a thorough description of the annotation process and annotator credentials (Appendix C).
>
> **Shorten method & data description sections + qualitative visualizations** We understand that each reader will have different editing preferences, with some sections to be preferably expanded and others shortened. We intentionally expand on data-related sections as this is a dataset paper, and we hope that the community will be able to replicate or improve our methodology to extend to other datasets or modalities. We have moved details for the gold standard annotation to supplementary material to expand on other sections that the reviewers were interested in. We have revised our paper to add more qualitative visualizations (Appendix G in supplementary).
>
> **Dataset evaluation expanded, how data collected can improve performance compared to existing data:** The authors have provided code in notebooks in the PhysioNet repository to calculate performance in detail for each type of relationship extracted in the dataset. Due to the large number of different types of annotations, we left the granular object-attribute and comparison relation evaluation in the PhysioNet commit (or there will be thousands of numbers reported). We point to these detailed evaluation results more clearly in our paper (lines 325-336). Since all other similar datasets are only labeled globally, there is no easy way to do apples-to-apples comparisons. Note that only a few open-sourced large CXR datasets exist: MIMIC-CXR, Stanford CheXpert, NIH, with global *not localized* labels as our dataset. We are not able to run and evaluate our pipeline on these other datasets since their CXR reports are not available due to HIPPA concerns. However, we are in discussion with dataset sources to expand the Chest ImaGenome work in the future.
>
> **Example annotated CXR regions:** We revised our paper accordingly (Figure 2)
>
> **What is silver dataset and how is it different from the Chest ImaGenome dataset?** As explained in the Methods and Data Description sections, for example in lines 188-190, etc., the term silver dataset refers to the automatically generated/labeled data, while gold refers to the human-validated-and-corrected subset that the annotators have labeled so as to verify the quality of the silver labels. The gold dataset is currently too small for training (or testing for rarer labels), hence it is advisable to use the large-scale silver data for model development. We have provided the splits we used in our experiment so that others can use the same split for reporting in the future.
>
> **Explain use of existing data. It is not clear exactly how and whether any permissions should be sought for this:** In our paper, we explain that the images come from the original MIMIC dataset and that in collaboration with the original MIMIC team, we add value with the construction of the Chest ImaGenome dataset. Additionally, Dr. Leo Celi (Chest ImaGenome co-author) is the PI of the original MIMIC work. We have added a paragraph (lines 89-95) to explicitly state the origin of the data, in addition to informed consent and HIPAA training requirements.

---

> > ### Comment · Reviewer_xHyG · 2021-07-21
> > **Responses**
> >
> > Thank you for the clarifications and additional images. I was able to see the data and notebooks, and indeed they are convincing and seem to be ready for use by researchers. I have updated my rating.

---

### Official Review · Reviewer_9iDs · 2021-07-03
**Well done dataset construction and important motivation that will have impact on machine learning aided clinical reasoning.**

**Rating:** 8
**Confidence:** 4

**Strengths:**

This is an incredibly well done dataset / benchmark paper. First, the paper is clearly written and does a good job of walking the reader through the motivations of the dataset and how it was constructed. It would be very easy for a researcher to read this paper and understand how to use the dataset. The study followed a gold standard for collecting and constructing datasets in healthcare which is great to see. This can give researchers confidence when using the dataset in their research. This paper is one of the first to provide a visual genome for chest x-rays which will be very valuable to the ML4H community.


**Weaknesses:**

The exposition on the example tasks could be motivated better. The creation of the dataset is very well motivated but then the evaluation tasks chosen did not appear well motivated to me. This is important as the evaluation tasks may be seen by the community as benchmark tasks to improve upon. I would provide more detailed motivation for these tasks. Additionally, it would be great to see confidence intervals for the AUC scores. Why was AUC used as the metric for those tasks?

**Additional Feedback:**

I believe this dataset will have large impact on the advancement of methods built for aiding radiology with machine learning. The authors have done a great job in constructing this dataset. More exposition on clinically relevant tasks and empirical evaluations of these tasks would make the paper even stronger.

Small nit: Broken reference to Table 1 on Page 2.

**Clarity:**

The paper is very clear, I have no comments on improvements.


**Correctness:**

Everything is correct to the best of my knowledge.


**Documentation:**

A great amount of documentation is present for this dataset. I appreciate the creation of Jupyter notebooks to support the statements about the statistics for the dataset. Additionally, following the PhysioNet guidelines they is a large amount of supporting documentation.

**Ethics:**

There are no ethical concerns that are apparent.


**Relation To Prior Work:**

It is clear that the authors built directly on prior work in constructing visual genomes. This is indicated as the authors provide a table that is an analogue between their Chest ImaGenome and the Visual Genome. Additionally, it is clear how this dataset differs from the existing CXR datasets that are publicly available. Especially given that no other CXR dataset has scene graph data.


**Summary And Contributions:**

This study constructed a new chest x-ray dataset for understanding clinical reasoning for diagnosing commonly found illnesses in chest-xrays. Inspired by the Visual Genome efforts they created a scene graph dataset for chest x-rays to improve current efforts for machine learning aided clinical reasoning. The authors provide a dataset through PhysioNet and example prediction tasks that can be done using the dataset. It is the first visual genome style dataset for chest x-rays.

---

> ### Author Response · Authors · 2021-07-14
> **Thank you for the positive comments and feedback. Please find our responses below**
>
> **Example tasks detailed motivation & evaluation metrics:** The example tasks are chosen due to their importance in the medical imaging domain, i.e., common disease detection and localization as well as the temporal aspect of localized findings (monitoring of patient condition over time, predicting change between sequential CXR exams, etc.). We have revised our manuscript to expand the motivation for both tasks (lines 337-397). We chose AUC score for attribute localization (Example Task 2, i.e., correct attribute in correct location), as it has been well-established in previous related work, for example in CheXGCN as well as on CXR attributes global classification (AUC is the metric that is usually reported in such works). For Example Task 1, our evaluation metric is accuracy, since the data is more balanced.
>
> **More exposition on clinically relevant tasks and empirical evaluations:** Thank you for the suggestions. As this is a dataset paper, we focus more on explaining the resources in the dataset. The dataset has been constructed with multiple possible downstream tasks in mind. However, we picked two tasks that can have the most immediate clinical applications: (i) outputting both the anatomical location and the type of CXR attribute for an image (Example Task 2), and (ii) comparing whether an anatomical location has worsened or improved across sequential exams (Example Task 1). Clinically, these are the two most important tasks for radiologists to report when interpreting CXRs. For clinical application 2), we picked the most clinically interesting attributes with good training support as a baseline to show in the paper. Knowing the anatomical location of non-specific findings/attributes on CXR images can help with narrowing down possible disease diagnoses and guide the next steps in requesting more specific imaging exams or treatment. For clinical application 1), CXRs are commonly repeatedly requested in the clinical workflow to assess changes for a myriad of attributes. We restricted the problem to attributes related to congestive heart failure (CHF) disease for the comparison experiments as a baseline because fluid management for CHF is one of the most routine clinical tasks for which CXRs can be ordered to guide the next steps (e.g., whether to give more intravenous fluid or give diuretics, etc.). However, users of this dataset can also explore comparison changes for other CXR attributes (e.g., pneumonia). There is also plenty of room for technical model improvements using this dataset, which we think other researchers can make and aim to publish in technical conferences. The clinical applications can be the same or similar, but perhaps a different model architecture or problem formulation (e.g., link prediction using GNNs) can achieve better results.

---

### Official Review · Reviewer_Jdyk · 2021-07-04
**A large dataset hindered by generality and accuracy issues**

**Rating:** 4
**Confidence:** 1
**Clarity:** The paper is clear.

**Strengths:**

Images of chest x rays are an important diagnostic tool for medicine, and prior work has leveraged them to build classifiers that approximate the accuracy of a trained radiologist.

As the authors detail in Table 1, the dataset they have produced has many attributes that are missing from prior datasets, making this one a potentially unique resource.

**Weaknesses:**

It's unclear how to gain access to this dataset since the license that the authors link to is itself behind a login wall.

The dataset was gathered from emergency room patients. Does this imply that the dataset doesn't have negative training examples, i.e., x rays of clean lungs?

The demographics of the patient pool are unknown. Is it sufficiently representative of the population to be able to train unbiased models? Further, the sourcing of data from a single hospital also raises the specter of generality problems.

The object-object evaluation metrics in Table 3 are not great, which raises questions about downstream use of the data.

**Additional Feedback:**

I thank the authors for their comments.

**Correctness:**

As noted above, I have concerns about the accuracy of the data as derived from NLP algorithms.

**Documentation:**

The documentation looked good to me.

**Ethics:**

The paper doesn't say anything about informed consent of the patients (if it's there, I missed it). This was quite concerning to me.

**Relation To Prior Work:**

Table 1 was quite helpful to understanding this datasets' advantages over prior work.

**Summary And Contributions:**

This paper presents a dataset of chest x rays that has been annotated based on medical records. This dataset is longitudinal per patient and includes hundreds of different medical conditions, positioning it favorably against similar datasets from prior work.

---

> ### Author Response · Authors · 2021-07-14
> **We thank the reviewer for the comments. Please find our responses below**
>
> **Access** The password for the review dataset version was supplied on OpenReview and was supposed to be passed to reviewers. The dataset is on PhysioNet (https://doi.org/10.13026/wv01-y230), hence can be downloaded by any MIMIC credentialed researcher. Note that our dataset is in the form of supplement to MIMIC-CXR and original images need to be collected from the MIMIC-CXR PhysioNet repository.
>
> **Negative examples, i.e., x-rays of clean lungs** The emergency room captures a broad spectrum of patients, many of whom will have negative CXRs, i.e., collected data also contain normal CXRs. We have labeled CXRs anatomically for a large number of important clinical abnormal and normal findings. In fact, in this dataset, there are specific “lung (object) is (relation) normal (attribute)” annotations, and many more (there are up to 1256 types of relations between different objects and attributes).
>
> **Patient demographics**  Demographics are well described in the original MIMIC-CXR dataset, from which Chest Imagenome dataset is derived. MIMIC-CXR was collected from an ED patient population in the northeast US. The source hospital system (Beth Israel Deaconess Medical Center, BIDMC) is academic. BIDMC gets referrals from a wide geographic region and looks after complex tertiary patients and many safety-net community patient populations.
>
> **Bias/generality** Only MIMIC-CXR is large enough, publicly available and has both CXRs and good quality text reports available to conduct our Chest ImaGenome work. Generalization and model biases are always a problem, even if the dataset came from multiple hospital systems. There are seasonal, X-ray machine-level, geographical data drifts, and unpredictable events, e.g., pandemics, that may change disease distribution and clinical CXR request practices. MIMIC-CXR is collected from an ED population. The ED population is the least specialized/filtered population in medicine if one is worried about disease biases. MIMIC-CXR has all CXRs from both inpatient and outpatient departments that ED patients also presented to during the 2-year data collection period. Underrepresented populations and diseases (some diseases are just much rarer) are inherent problems in healthcare data that require their own technical innovations to address (i.e., outside the scope of our work). Our goal is NOT to provide a dataset that guarantees training generalizable models with no biases, there are just too many levels of technical and practical complexities involved (though we certainly hope the dataset would help assess for some of these issues). Our goal is to 1) show a potentially scalable method to graphically and locally describe a medical imaging modality that makes sense clinically, 2) aid in more quantitative evaluation of explainability in the CXR space, and 3) spur downstream clinically/technically interesting research. By structuring the dataset graphically in such amount of detail, we are indirectly allowing researchers to explore some CXR model bias issues, e.g., whether models predict the most common location for a given abnormal finding and not really looking at features. There are already some techniques in the Visual Dialogue literature to explore this example issue. This points to another interesting added value that Chest Imagenome provides for the research community.
>
> **Evaluation Table3** Automatically localizing attributes to anatomy via NLP is by far a solved technical problem. Even though the dataset is not perfect yet, we have shown an approach whereby one could semi-automatically derive richer labels from medical imaging data, which is important as otherwise the cost to annotate such a detailed dataset is prohibitive. Furthermore, the current benchmark for accuracy of automatically extracted image-level labels via NLP is not much better than the accuracy of our localized labels. If not labeling locally, our pipeline's performance would be all above 90% though that defeats the goal of our work. With this starting dataset, the research community can work towards improving NLP for localizing attributes to anatomical objects and localized comparison relations. There is also room to address missing links in the scene graph dataset with additional technical innovations, e.g., relations that are true for images but even the original CXR reports failed to describe them, as well as downstream clinical usecases in developing anatomically-centered image encoding models which can help build more clinically explainable models. These use cases are outside the scope of this current work (dataset construction+description) but were certainly the underlying intent for this dataset.
>
> **Informed consent** MIMIC data is de-identified and the institutional review boards of MIT (No. 0403000206) and Beth Israel Deaconess Medical Center (2001-P-001699/14) both approved the use for research. We added a paragraph (l.89-95) to explicitly state data origin, informed consent & HIPPA training requirements.

---

### Author Response · Authors · 2021-07-14
**Response to reviewers, revised based on reviewer feedback**

Dear reviewers and AC,

We thank the reviewers for the constructive feedback and suggestions that have substantially improved our manuscript. We believe we have addressed all concerns and suggestions. In addition, we have revised our paper according to the reviewers' suggestions.

More specifically, we summarize our revisions and explanations:
- We have included more details about annotator credentials and the annotation process.
- We have explained the added value that Chest ImaGenome brings to the MIMIC-CRX dataset, and in the medical imaging research community in general.
- We addressed concerns about the patient demographics/population and the patient informed consent.
- We have provided explanations to reviewers' questions, e.g., on the existence of negative training examples, evaluation metrics, etc.
- We have improved the Clinical Applications section to better motivate our example tasks.
- We added an example image with annotated CXR regions, as well as qualitative examples.
- We have tried to resolve dataset access problems. Please note that the password for the review version of the dataset was supplied on OpenReview and was supposed to be passed on to reviewers. The dataset is now been accepted on PhysioNet for public viewing (https://doi.org/10.13026/wv01-y230), hence can now be downloaded by any *MIMIC credentialed* researchers. Note that our dataset is in the form of a supplement to the MIMIC-CXR dataset and the original images need to be collected from the MIMIC-CXR PhysioNet repository.

Thank you again to all reviewers!

---

### Decision · Program_Chairs · 2021-07-27

**Decision:**

Reject

**Comment:**

The authors introduce a new chest x-ray dataset that scene graphs inspired by the Visual Genome project. The reviewers note novelty and scope of the dataset, as well as the significance of the task. However, they also raise concerns about data accessibility and the motivation for some of the design choices used. They acknowledge the author's rebuttal but nonetheless they recommend against accepting the paper at this point.